# Weakened western Indian Ocean dominance on Antarctic sea ice variability in a changing climate

Li Zhang [1,2,5] ✉, Xuya Ren[1,2,5], Wenju Cai [1,2,3] ✉, Xichen Li [4] & Lixin Wu [1,2]

Patterns of sea surface temperature (SST) anomalies of the Indian Ocean Dipole (IOD) exhibit strong diversity, ranging from being dominated by the western tropical Indian Ocean (WTIO) to the eastern tropical Indian Ocean (ETIO). Whether and how the different types of IOD variability patterns affect the variability of Antarctic sea ice is not known, nor is how the impact may change in a warming climate. Here, we find that the leading mode of austral spring Antarctic sea ice variability is dominated by WTIO SST variability rather than ETIO SST or El Niño–Southern Oscillation. WTIO warm SST anomalies excite a poleward-propagating Rossby wave, inducing a tri-polar anomaly pattern characterized by a decrease in sea ice near the Amundsen Sea but an increase in regions on both sides. Such impact has been weakening in the two decades post-2000, accompanied by weakened WTIO SST variability. Under greenhouse warming, climate models project a decrease in WTIO SST variability, suggesting that the reduced impact on Antarctic sea ice from the IOD will likely to continue, facilitating a fast decline of Antarctic sea ice.

The Indian Ocean dipole mode (IOD, refs. 1,2) refers to variability of the equatorial zonal sea surface temperature (SST) variability in the tropical Indian Ocean and is defined as the difference in area mean SST anomalies between the western pole (10° S–10° N, 50° E–70° E) and the eastern pole (10°S–0°, 90° E–110° E), known as a dipole mode index[1,2] (DMI). The IOD is known to influence weather and climate in many tropical and extratropical regions[3]. During a positive IOD, anomalous cooling in the east drives anomalous sinking motion over the eastern tropical Indian Ocean, and anomalous warming in the west leads to anomalous ascending motion there. Similar to anomalies of El Niño–Southern Oscillation (ENSO), the associated convective anomalies generate anomalous convective diabatic heating generating a Rossby wave source[4–8]. The associated Rossby wave trains curve poleward and eastward towards Antarctica with alternating centers of high and low-pressure anomalies.

Through such atmosphere teleconnections, the anomalous ocean-atmospheric circulations during positive IOD events affect weather and climate in remote regions, including droughts, heat waves, and bushfires in Australia[9–12], but floods[13,14] and malaria outbreaks in East Africa[15], and coral reef death across western Sumatra[16]. Around the Antarctic, sea ice variability is affected because of wind-driven dynamic and thermodynamic responses to the anomalous atmospheric circulation[5–7,17]. However, the IOD impact on Southern Hemisphere sea ice variability was examined in the presence of concurrent SST of ENSO, which generates similar teleconnections[18–20]. As such, the IOD impact is often regarded as secondary to, or reinforcing, that of ENSO[8,18], although such IOD-induced wave trains are generated with ENSO-related SST anomalies removed[21]. More importantly, using the DMI, there is an implicit assumption of no diversity in the IOD anomaly pattern. Therefore, whether the IOD impact differs with different IOD anomaly patterns is unknown.

Positive IOD events display different flavors[22–24]. One such manifestation shows two positive IOD regimes, depicted by two separate indices defined as the strong IOD events with strong eastern tropical

[1]Frontiers Science Center for Deep Ocean Multispheres and Earth System and Key Laboratory of Physical Oceanography/Academy of the Future Ocean, Ocean University of China, Qingdao, China. [2]Laoshan Laboratory, Qingdao, China. [3]CSIRO Oceans and Atmosphere Flagship, Aspendale, VIC 3195, Australia. [4]Institute of Atmospheric Physics, Chinese Academy of Sciences, Beijing 100029, China. [5]These authors contributed equally: Li Zhang, Xuya Ren. ✉e-mail: zhangli@ouc.edu.cn; Wenju.Cai@csiro.au

Indian Ocean (ETIO) cool anomalies, and moderate IOD events dominated by western warm anomalies, respectively[25]. Although tropical convection anomalies in the western region have co-varying anomalies with those in the eastern region, and vice versa, differences in the relative strength of convective anomalies in the two centers result in vastly different wave train patterns[8]. Thus, SST variability dominated by the western tropical Indian Ocean (WTIO) and ETIO could induce vastly different atmospheric teleconnection patterns with different impacts on the Antarctic climate.

Using observational data and numerical simulation, here we find that the leading mode of Antarctic sea ice in austral spring, a tri-polar anomaly pattern, is predominantly influenced by WTIO SST anomalies, and is distinctively different from that induced by ENSO. Further, this WTIO dominance of the Indo-Pacific influence on Antarctic sea ice has been weakening, the cause of which is unknown, but is in line with a projected change in the IOD over the 21st century.

## Results

### The tri-polar sea ice pattern dominated by the WTIO SST

An IOD starts to develop in austral winter and usually peaks in austral spring, with a seasonal dependence on its atmospheric teleconnections. We examine the influence of SST over the tropical Indian Ocean (20° S–20° N, 50° E–110° E) on Antarctic sea ice concentration (SIC) from austral winter (June–August) to early austral summer (November–January) using a maximum covariance analysis (MCA). To extract the "pure" tropical Indian Ocean influence on Antarctic sea ice, we remove ENSO signals using a linear regression (see "Observational data" and "Maximum covariance analysis" in "Methods" section).

The squared covariance of the first MCA mode shows a peak in August-September-October (ASO) (Fig. 1a), with a squared covariance fraction of 56%. The corresponding time series of the first MCA-SST pattern reflects fluctuations of the IOD ($r = 0.44$; Fig. 1b, upper), and the first MCA pattern shows a broad warming across the western and central tropical Indian Ocean and a weak cooling center over the ETIO (Fig. 1c). For the sea ice, the MCA-SIC pattern (Fig. 1d) manifests as an Antarctic tri-polar structure, with sea ice retreating in the Amundsen Sea while expanding near the Antarctic Peninsula and west of the Ross Sea. This tri-polar sea ice pattern resembles ($r = 0.98$) the first empirical orthogonal function (EOF1) spatial pattern of Antarctic ASO SIC (Supplementary Fig. 1a), accounting for 24% of the total variance. Furthermore, the grid-point correlation coefficients between the principal component time series of the SIC EOF1 and SST reveal the IOD-like pattern with the most significant region in the WTIO (Supplementary Fig. 1b), reinforcing the MCA results.

Using the DMI determined from anomalous SST gradient between the western pole and the eastern pole assumes that the IOD possesses little pattern diversity, making it difficult to identify the associated impacts of each pole. To this end, we consider time series in the WTIO (10° S–10° N, 50°–70° E) and the ETIO (10° S–0°, 90° E–110° E) alongside the overall tropical Indian Ocean (Fig. 1b, lower). The time series of the WTIO and ETIO SST anomalies are largely independent, with no statistically significant linear ($r = 0.05$) or nonlinear ($\alpha = 0.08$) relationship. As expected, the time series of WTIO ($r = 0.94$), rather than that of ETIO ($r = 0.23$), largely represents the first MCA-SST time series, whose corresponding SST pattern is dominated by the WTIO.

Previous studies used the first two principal components of an EOF analysis to construct moderate IOD and strong IOD (see "Strong

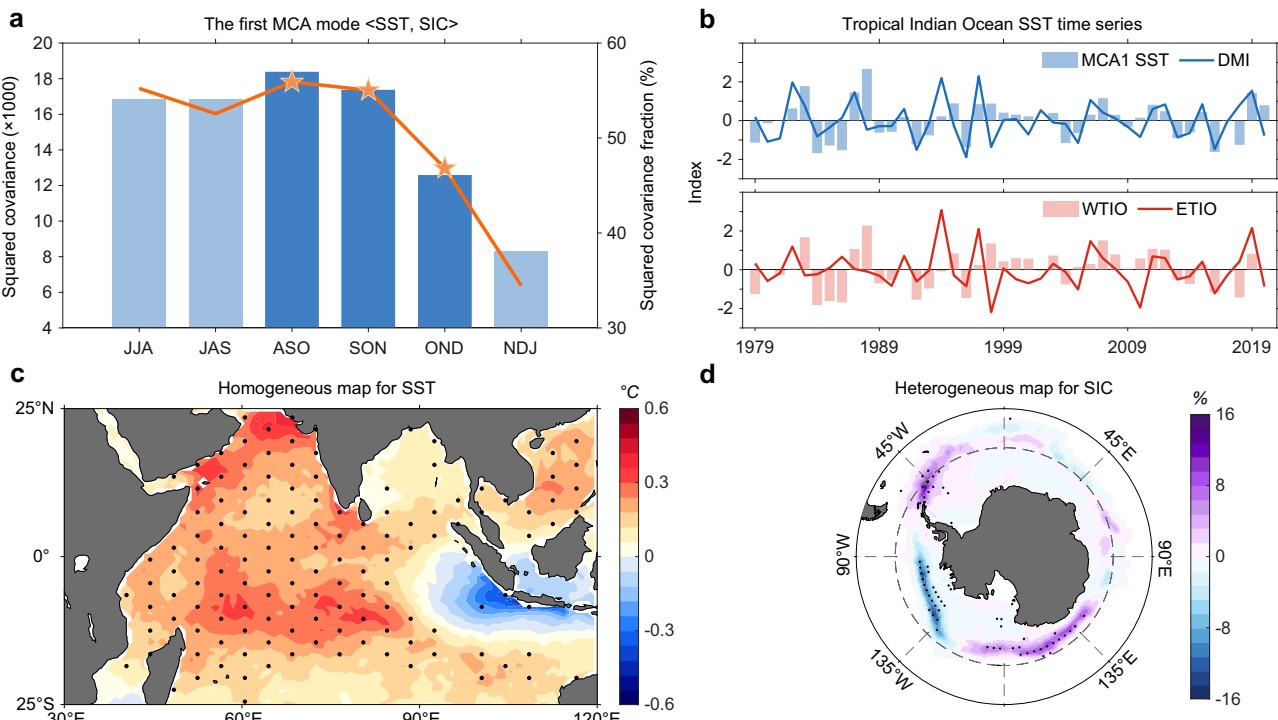

**Fig. 1 | The tri-pole pattern of observed sea ice dominated by western tropical Indian Ocean in austral spring. a** Seasonal squared covariance (blue bar; ×1000) and squared covariance fraction (orange line; %) of the first maximum covariance analysis (MCA1; see "Maximum covariance analysis" in "Methods" section) mode between tropical Indian Ocean sea surface temperature (SST; 20° S–20° N, 50° E–110° E) and Antarctic sea ice concentration (SIC; 60° S–90° S) for the 1979–2020 period. Darker blue bar and orange pentagram indicate the 95% confidence level based on the Monte Carlo test, respectively. **b** Normalized time series of MCA1 SST (blue bar), dipole mode index (DMI; blue line), western tropical Indian Ocean (WTIO; red bar; 10° S–10° N, 50° E–70° E), and eastern tropical Indian Ocean (ETIO; red line; 10° S–0°, 90° E–110° E) for August-September-October (ASO). **c, d** The MCA1 mode of **c** SST (°C) and **d** SIC (%) for the ASO 1979–2020 period. Stippling indicates the 95% confidence level based on the t-test. The leading mode of austral spring Antarctic sea ice variability is dominated by WTIO SST variability rather than ETIO SST or El Niño-Southern Oscillation (ENSO). Source data are provided as a Source Data file.

positive IOD and moderate positive IOD" in "Methods" section), dominated by the WTIO and the ETIO, respectively[25,26] (Supplementary Fig. 2). These analyses find a nonlinear relationship between the first two principal components that distinguish the WTIO- and ETIO-dominated IOD events. The WTIO and ETIO SST time series show strong correlations with the moderate IOD and strong IOD indices, at 0.71 and 0.91, respectively (Supplementary Fig. 2). The DMI primarily represents ETIO-dominated IOD events, as reflected in its strong correlation with the ETIO ($r = 0.84$) or strong IOD ($r = 0.86$). However, the relationship of the DMI with the WTIO ($r = 0.22$) or with moderate IOD ($r = 0.32$) is much weaker. Thus, DMI might not be effective for identifying the impact from the WTIO.

Regression of austral spring grid-point SST and SIC anomalies onto the normalized WTIO SST time series (Supplementary Fig. 3a, b) displays similar patterns to those in the MCA results. However, the ETIO- or DMI-induced SIC anomalies display a weaker and statistically insignificant pattern, although the SST anomaly amplitude associated with the ETIO or the DMI is larger than that of the WTIO (Supplementary Fig. 3c–f). The WTIO dominance on the tri-polar SIC anomalies persists into September-October-November (not shown). Thus, WTIO SST variability predominantly affects the principal pattern of austral spring Antarctic SIC variability, which at a positive phase features a decrease in sea ice near the Amundsen Sea, but an increase in regions on both sides.

## Weakened influence by the WTIO SST after the late 1990s

Since 2000, the dominant influence of the WTIO SST on Antarctic sea ice has weakened, as evidenced by the consistent results from the sliding correlations between MCA-SST, WTIO, and tri-polar sea ice anomalies (Fig. 2a). Taking the WTIO SST as an example, a 21-year sliding correlation with the tri-polar sea ice anomalies shows a sharp reduction around 1999, from a value of 0.64 that exceeds the 99% confidence level during the pre-1999 period to a low value of 0.21 during the post-1999 period (see "Sliding correlation and sliding standard deviation" in "Methods" section). This is insensitive to sliding window lengths (e.g., 17-year or 19-year; Supplementary Fig. 4), suggesting a robust change in the WTIO-Antarctic sea ice relationship.

WTIO SST variability exhibits a pronounced and coherent reduction during the satellite era, with a decline trend of −0.208 standard deviation decade$^{-1}$ averaged across six reanalysis products (Fig. 2b; see "Observational data" and "Sliding correlation and sliding standard deviation" in "Methods" section), which is also reflected in the SST anomaly difference between the pre-1999 and the post-1999 periods (Fig. 2c). During the pre-1999 period, positive SST anomalies were seen throughout the tropical Indian Ocean except for some small negative values near Indonesia. However, the amplitude of the positive WTIO anomalies decreased and the negative anomalies in the ETIO were more prominent during the post-1999 period, thus forming a clearer SST zonal dipole structure. During the pre-1999 period, SIC anomalies associated with the WTIO exhibit a well-defined tri-polar pattern, whereas the SIC anomalies are weak during the post-1999 period (Fig. 2d). Below we examine the mechanism for the pre-1999 influence on Antarctic sea ice by the WTIO SST anomalies and discuss plausible causes for the post-1999 decrease in the impact.

## The mechanism whereby WTIO anomalies influence Antarctic sea ice

Because the influence of the WTIO SST is stronger in the pre-1999 period, we use this period to understand the teleconnection mechanism. According to the Rossby wave theory, the SST anomalies lead to changes in tropical convection driving an anomalous divergence in the upper troposphere. The divergent flow, in turn, excites Rossby wave sources in the subtropics and forces Rossby wave trains that propagate from the region of the tropical anomalies to the extratropics[27,28], ultimately reaching the Antarctic less in two weeks[29].

Along with the increased convective precipitation, latent heat release from ascending motion leads to anomalous divergent winds over the WTIO, promoting convergent motion across the ETIO (Fig. 3a, left). The Rossby wave source can be generated by anomalous advection of the mean meridional gradient of absolute vorticity.

There is a conspicuously anomalous positive Rossby wave source off the southwest of Australia (Fig. 3a, right; see "Rossby wave source and wave activity flux" in "Methods" section) in association with the divergent flow from the eastern subtropical Indian Ocean. Rossby wave activity flux, commonly used to inspect the propagation and to track the group velocity of the Rossby wave train[30,31], indicates that the Rossby wave source over the southwest of Australia serves as the source of the wave train arcing poleward and eastward into the mid- and high-latitudes (Fig. 3b, vectors). This wave train is characterized by alternating positive and negative 200 hPa geopotential height (Z200) anomalies (Fig. 3b, shading; R1). Divergence of wave activity flux over South Africa and Z200 anomalies over the south Indian Ocean (65° S–15° S) indicates an additional source of the Rossby wave train (R2).

The synoptic high-frequency transient eddy activity along with its dynamic forcing contributes to different locations of the teleconnection lobes at mid- to high-latitudes[32,33]. The storm track-induced geopotential height tendency[34–36] ($Z_{tend}$; Fig. 3c, shading, and Supplementary Fig. 5; see "Eddy-induced geopotential height tendency" in "Methods" section) exhibits a barotropic response throughout the troposphere, dominated by anomalous eddy vorticity forcing (Supplementary Fig. 6). Large $Z_{tend}$ anomalies appear along ~70° S–30° S and reinforce the anomalous geopotential height there, leading to the vertical barotropic nature in the extratropics (Fig. 3c, contours).

Geostrophic balance connects sea level pressure (SLP) to surface wind anomalies, which affect regional-scale anomalies in SIC and surface air temperature (SAT) (refs. 7,29,37,38). In the high latitudes, high and low-pressure anomalies over ~120° E–0° induce anticyclonic and cyclonic circulations, respectively (Fig. 3d, shading). The associated anomalous northerlies and southerlies drive anomalous warm-air and cold-air advection (Fig. 3d, contours), decreasing and increasing SIC, respectively, leading to the tri-polar distribution. These changes are accompanied by the downward longwave radiation induced by water vapor anomalies around Antarctica (Supplementary Fig. 7). Such atmospheric bridge can be seen using data over the entire period (1979–2020), but weaker than during the pre-1999 period (Supplementary Fig. 8a, b). Moreover, the ETIO anomalies excite a distinct but weak wave train that emanates from southwest of Australia and propagates along the zonal direction. This wave train originates from the eastward-moving Rossby wave source and cannot drive the tri-polar structure (Supplementary Fig. 8c, d). During the post-1999 period, a weaker positive Rossby wave source anomaly emerges around 75° E–100° E, which is farther west relative to its location during the pre-1999 period (Supplementary Fig. 8e). This Rossby wave source is similarly associated with small and statistically insignificant atmospheric responses in mid- and high-latitudes, indicating a modest impact on sea ice (Supplementary Fig. 8e, f).

There is no consensus on the IOD influence on Antarctic sea ice using the DMI, which is dominated by the ETIO SST anomalies. On the one hand, because the IOD often co-occurs with ENSO, the influence of the IOD might reflect that by ENSO, with the IOD impact on sea ice considered secondary. On the other hand, the influence of the IOD alone was suggested to be unable to reach the Bellingshausen Sea, Antarctic Peninsula region, or Weddell Sea[18].

To further clarify the role played by ENSO, we analyze ENSO-related teleconnection separately and find it to be vastly different from that induced by the WTIO. During austral spring, teleconnection generated by ENSO shows a Pacific–South American (PSA)-like wave train[39,40], affecting the Amundsen Sea Low that alters the Antarctic

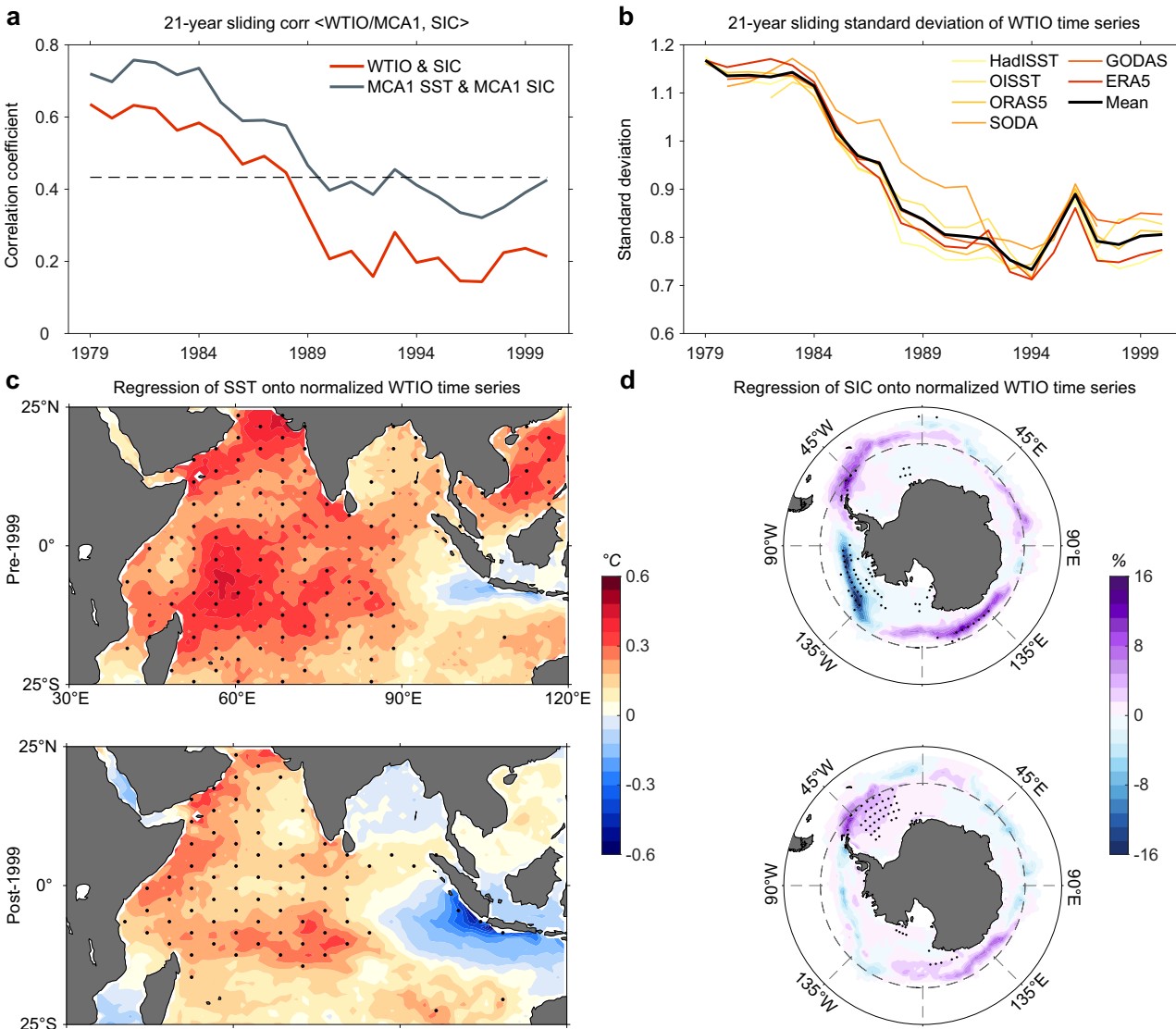

**Fig. 2 | Weakened western tropical Indian Ocean variability and its impacts on Antarctic sea ice after 2000. a** 21-year sliding correlation coefficients between the western tropical Indian Ocean (WTIO) time series and the first principal component of empirical orthogonal function (EOF) of Antarctic sea ice concentration (SIC; red line), as well as the first maximum covariance analysis (MCA1) sea surface temperature (SST) and SIC time series (gray line). The dashed line indicates the 95% confidence level based on the *t*-test. **b** 21-year sliding standard deviation of the WTIO time series using six reanalysis products and the mean values across them (see "Data" and "Sliding correlation and sliding standard deviation" in "Methods"

section). The decline trend is −0.208 standard deviation decade⁻¹ average across these six reanalysis products. **c** Regression of SST (°C) onto the normalized WTIO time series for the pre-1999 (upper) and post-1999 (lower) periods. **d** Same as (**c**) but of SIC (%). The x-axis in (**a**, **b**) indicates the starting year in the 21-year sliding window (e.g., 1979 indicates the correlation coefficient or standard deviation for the 1979 – 1999 period). Stippling indicates the 95% confidence level based on the *t*-test. The impact of WTIO SST anomalies on Antarctic sea ice has been weakening in the two decades post-2000, accompanied by weakened WTIO SST variability. Source data are provided as a Source Data file.

dipole structure of SIC anomalies (Supplementary Fig. 9). This SIC dipole appears as the SIC EOF2, independent and distinct from tri-polar SIC anomalies dominated by the WTIO, and is related to convective anomalies over the tropical Pacific (Supplementary Fig. 10).

### Numerical model experiments simulate WTIO-associated teleconnection

To test our hypothesis of a direct causal link between the WTIO SST anomalies and the Antarctic atmospheric pattern, we perform a set of atmospheric general circulation model (AGCM) experiments using the Community Atmosphere Model version 5 (CAM5), a state-of-the-art atmospheric model (see "CAM5 model simulations" in "Methods" section). Here, we apply the climatological annual cycle of SST derived from the Hadley Centre sea ice and SST dataset (HadISST; 1871–2008)

as the boundary forcing for the control simulation (CTRL). In the sensitivity experiment, an anomalous SST pattern is added to the climatology SST in ASO in the tropical Indian Ocean region, while other setups remain the same as the CTRL. The SST anomalies forcing are imposed following the WTIO SST pattern without amplitude scaling (i.e., Fig. 2c). All experiments are integrated for 40 years, and the atmospheric response is taken as a 30-member ensemble-mean difference between the sensitivity run and the control run. The simulated Z200 response (Fig. 4a) to a WTIO warming reproduces the results. The model also reproduces a tri-polar response of SAT in the Antarctic, albeit the SLP anomalies near the Amundsen Sea shift slightly eastward (Fig. 4b). The realistic simulation is despite no air−sea-ice−ocean interactions in our atmosphere-only model, which if present, would facilitate a more realistic pattern because SAT anomalies in the polar

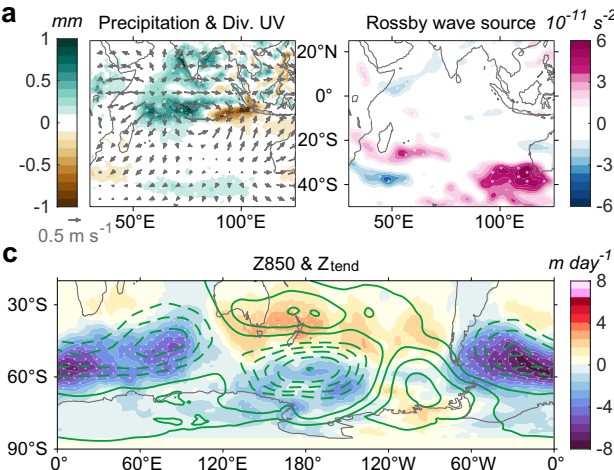

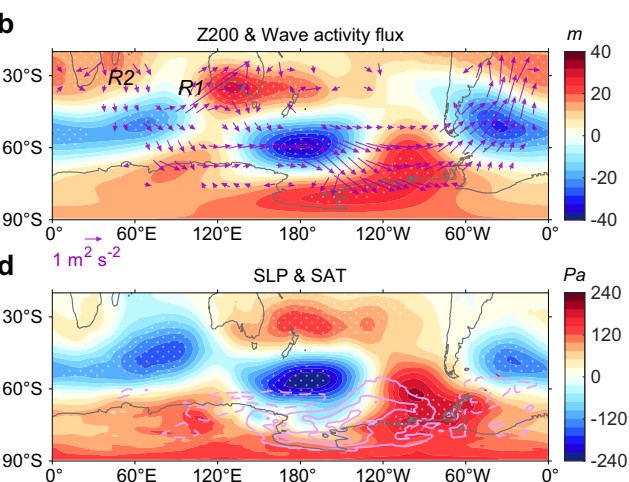

**Fig. 3 | Pre-1999 atmospheric bridge associated with the western tropical Indian Ocean. a–d** Regressions of **a** convective precipitation (shading; mm), 200 hPa divergent wind (vectors; m s⁻¹), Rossby wave source (10⁻¹¹ s⁻²), **b** 200 hPa geopotential height (Z200; shading; m), wave activity flux (vectors; m² s⁻²; see "Rossby wave source and wave activity flux" in "Methods" section), **c** 850 hPa geopotential height tendency (Z$_{tend}$) induced by the sum of transient eddy heat flux and eddy vorticity flux (shading; m day⁻¹; see "Eddy-induced geopotential height tendency" in "Methods" section), 850 hPa geopotential height (Z850; green contours; positive solid and negative dashed; zero line omitted; starts from ± 4 m and interval ± 4 m), **d** sea level pressure (SLP; shading; Pa), and surface air temperature (SAT; light pink contours; positive solid and negative dashed; zero line omitted; starts from ± 0.37 °C and interval ± 0.5 °C) onto the normalized western tropical Indian Ocean (WTIO) time series for the pre-1999 period. "R1" and "R2" in (**b**) indicate the location of the two effective Rossby wave sources. Stippling indicates the 95% confidence level based on the *t*-test. The WTIO sea surface temperature (SST) variability promotes stationary Rossby wave patterns to the Southern Hemisphere high latitudes, reinforced by storm track activities. Atmospheric pressure anomalies, in turn, drive the tri-polar SAT and sea ice concentration (SIC) anomalies. Source data are provided as a Source Data file.

region would be amplified by the SIC response through the albedo feedback.

To provide further evidence for the influence by the WTIO alone, we further analyze a 10-member ensemble of Indian Ocean pacemaker experiments, in which time-evolving tropical Indian Ocean SST anomalies are nudged to observations and the rest of the model's coupled climate system free to evolve (see "Indian Ocean pacemaker experiments" in "Methods" section). Thus, the ensemble-mean of the pacemaker experiments isolates the influence from the WTIO SST anomalies. Consistent with the reanalysis and the AGCM results, the pacemaker experiments simulate anomalous Rossby wave trains, shown as Z200 and SLP alternative high and low anomaly centers in the mid- and high-latitudes of the Southern Hemisphere, affecting regional-scale SIC and SAT anomalies around the Antarctic (Fig. 4c, d). The coupled model experiments support our finding that the WTIO-generated atmospheric teleconnection dominates tri-polar sea ice anomalies. Thus, three independent methods, that is, reanalysis identification, atmospheric model experiments, and Indian Ocean pacemaker experiments, provide multiple lines of evidence for our mechanism.

## Weakened WTIO influence consistent with a decreased impact under greenhouse warming

The extent to which the post-1999 reduction in the WITO influence on Antarctic sea ice is driven by decadal variability or greenhouse warming cannot be addressed using observations alone, given the short period over which reliable observations are available. However, the observed difference in SST anomalies between the two periods is consistent with what is expected from a difference in the mean state change. Post-1999, the troposphere warms faster than the surface (Supplementary Fig. 11a), such that the response of equatorial easterlies to SST anomalies weakens[41]. Ekman pumping and zonal advection, the main forcing for warm anomalies in the WTIO, weakens, causing decreased WTIO variability[25]. The mean SST warming in the central tropical Indian Ocean leads to winds converging towards it, facilitating anomalous southeasterlies that extend to the equatorial

central and eastern Indian Ocean. These changes lead to an enhanced equatorial nonlinear zonal and vertical advection, in turn conducive to cold anomalies[1,2,42] in the ETIO (Supplementary Fig. 11b).

To gauge the likelihood that the post-1999 weakening in WTIO SST variability is partly due to climate change, we examine outputs from 45 models participating in the Coupled Model Intercomparison Project phase 6 (CMIP6). These models are forced with historical forcing until 2014 and with Shared Socioeconomic Pathways (SSP) 5–8.5 (SSP5–8.5) emission scenario from 2015 onward to 2099 (see "CMIP6 model simulations" in "Methods" section). ASO-averaged SST anomalies are calculated, referenced to the climatological mean over the whole period, and quadratically detrended. The WTIO time series is calculated from the observed pole.

We first compare simulated WTIO SST variability in the 21-year windows of the pre-1999 and post-1999 periods. A total of 31 out of 45 models (69%) simulate decreased variability of SST in the WTIO (Fig. 5a). The ensemble-mean shows a decrease of approximately 10% for SST variability in the WTIO, statistically significant above the 95% confidence according to a Bootstrap test (see "Bootstrap test" in "Methods" section). To assess the contribution of greenhouse warming during satellite era, we calculate 21-year sliding standard deviation by using the selected 31 models. The rate of decrease in WTIO SST variability in multi-model mean is −0.093 standard deviation decade⁻¹ (Supplementary Fig. 12), indicating that greenhouse warming contributes 45% to the observational decline (i.e., Fig. 2b, −0.208 standard deviation decade⁻¹). We further analyze differences in 21-year mean atmospheric temperature between the pre-1999 and post-1999 periods. The climate models reproduce the result that during the post-1999 period, the troposphere warms faster than the surface, consistent with the observed result (Supplementary Fig. 13a). Taken together, the results above suggest that the decrese in WTIO SST variability is likely in part forced by greenhouse warming.

We also compare WTIO SST variability between the 20th (1900–1999) and the 21st (2000–2099) centuries; a total of 37 out of 45 models (82%) show a decrease in WTIO SST variability in the 21st century (Fig. 5b). The strong intermodel consensus is supported by a

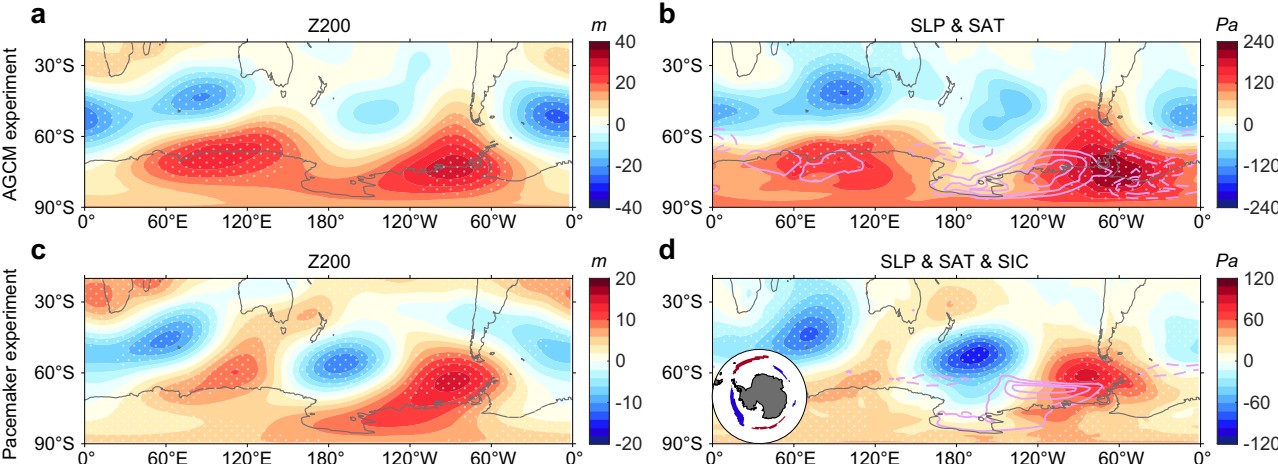

**Fig. 4 | Numerical model experiments assessing western tropical Indian Ocean sea surface temperature anomalies effects. a, b** Simulated **a** 200 hPa geopotential height (Z200; m), **b** sea level pressure (SLP; shading; Pa), and surface air temperature (SAT; light pink contours; positive solid and negative dashed; zero line omitted; starts from ± 0.3 °C and interval ± 0.25 °C) responses to western tropical Indian Ocean (WTIO) sea surface temperature (SST) forcing (i.e., Fig. 2c, pre-1999). **c, d** Regressions of **c** Z200 (m), **d** SLP (shading; Pa), SAT (light pink contours; positive solid and negative dashed; zero line omitted; starts from ± 0.3 °C and

interval ± 0.3 °C), and sea ice concentration (SIC; left bottom corner; positive red and negative blue) onto the normalized WTIO time series in the Indian Ocean pacemaker experiments for the pre-1999 period. See "Indian Ocean pacemaker experiments simulations" in "Methods" section. Stippling indicates the 95% confidence level based on the t-test. The Community Atmosphere Model version 5 (CAM5) simulations and Indian Ocean pacemaker experiments display a consistent atmospheric bridge, which is similar to the reanalysis result. Source data are provided as a Source Data file.

decrease of 13%, statistically significant above the 95% confidence according to a Bootstrap test. The long-term decrease in WTIO SST variability is similarly attributed to the rising tropospheric temperatures under greenhouse warming (Supplementary Fig. 13b). To illustrate the intermodel performance of SST anomalies associated with the WTIO, we perform a regression analysis (Supplementary Fig. 14), which shows a similar SST dipole structure dominated by the WTIO, but with a weak amplitude in the 21st century. The decrease is located in the western-central tropical Indian Ocean, with corresponding differences in atmospheric teleconnections associated with WTIO SST variability between the present-day and the future climate. The climate models reproduce atmospheric circulation anomalies in the mid-latitudes that are similar to the observed and numerical simulation results (Supplementary Fig. 15).

## Discussion

Our discovery that the leading mode of variability of austral spring Antarctic sea ice is a tri-polar pattern dominated by WTIO SST variability is robust. That WTIO SST variability dominates the influence of the Indo-Pacific on austral spring Antarctic sea ice variability is in contrast to a previous assumption of ENSO dominance, the impact of which, as we find, ranks the second and is of a different pattern (Supplementary Fig. 16). Western warm SST anomalies excite a poleward-extending Rossby wave train, inducing the tri-polar anomaly pattern characterized by a decrease in sea ice near the Amundsen Sea but an increase to regions both sides. Further, during the second half of the satellite era, the influence of the WTIO SST has weakened, with correspondingly decreased variability of the WTIO SST. Due to limited observations, we are unable to attribute the cause for the observed post-1999 weakening based on observations alone[43], but the weakening in WTIO SST variability and the resultant decrease in the impact on austral spring Antarctic sea ice is consistent with what is expected from greenhouse warming, with a strong intermodel agreement on reduced WTIO SST variability. Our result suggests that with a reduced inter-annual influence from the Indian Ocean projected to continue, the melt of Antarctic sea ice could accelerate as the signal of the impact from greenhouse warming becomes stronger relative to decreased inter-annual variability.

## Methods

### Observational data

Monthly and daily mean reanalysis data we used in this study come from the fifth generation European Center for Medium Range Weather Forecasts (ECMWF) reanalysis (ERA5) with a 0.25° horizontal resolution and 37 vertical levels in the atmosphere components[44]. We regridded the above data sets to a common 1° × 1° latitude–longitude grid before diagnosing. Anomalies are constructed with reference to the mean of the period 1979–2020, and then detrended. We remove the impacts of ENSO through a linear regression approach[45]. As the Niño3.4 index denotes the tropical Pacific signals, the new filed $\xi$ is derived from this formula: $\xi = \xi^* - Ni\tilde{n}o3.4 \times \frac{\text{cov}(\xi^*, Ni\tilde{n}o3.4)}{\text{var}(Ni\tilde{n}o3.4)}$, where $\xi^*$ denotes the original filed. This technique is used through our reanalysis study, except for Supplementary Figs. 1, 10, and 11. The Niño3.4 index is defined as the area-averaged SST anomalies in the range of 5° S–5° N, 170° E–120° W with a 5-month running mean employed[46], which is downloaded from https://psl.noaa.gov/gcos_wgsp/Timeseries/Nino34. The DMI index we used in our study is defined as the SST anomalies gradient between the western equatorial Indian Ocean (10° S–10° N and 50° E–70° E) and the south eastern equatorial Indian Ocean (10° S–0° N and 90° E–110° E)[46], which is downloaded from https://psl.noaa.gov/gcos_wgsp/Timeseries/DMI/.

We also add five other SST reanalysis products to repeat the 21-year sliding correlation of WTIO time series, and a multiproduct average of these total six products is then calculated. The five products are:

- HadISST (Hadley Centre Sea Ice and Sea Surface Temperature dataset from 1979 to 2020) (ref. 46);
- OISST v2 (NOAA Optimum Interpolation SST version 2 from 1982 to 2020) (ref. 47);
- ORA-S5 (ECMWF Ocean Reanalysis System 5 from 1979 to 2020) (ref. 48);
- SODA3.12.2 (Simple Ocean Data Assimilation version 3.12.2 from 1980 to 2017) (ref. 49);
- GODAS (NCEP Global Ocean Data Assimilation System from 1980 to 2020) (ref. 50).

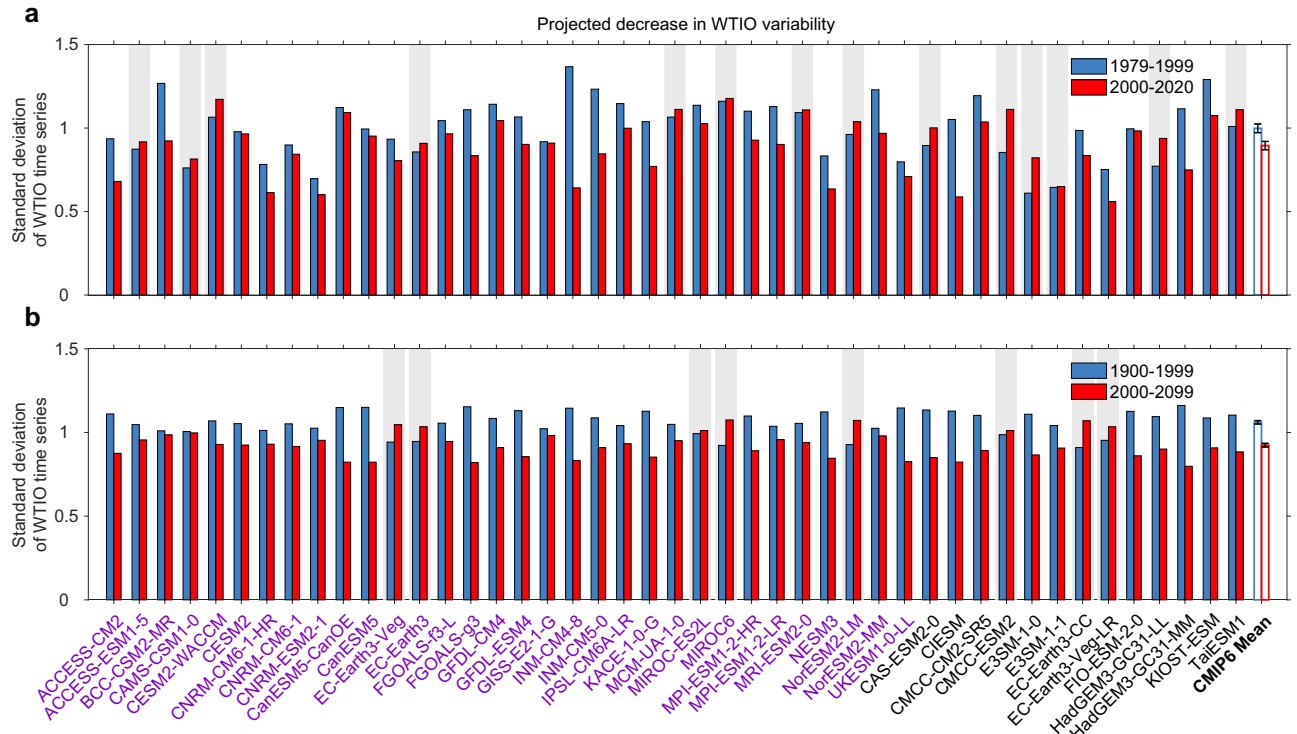

**Fig. 5 | Weakened western tropical Indian Ocean variability under greenhouse warming. a** A total of 31 out of the 45 models (69%) generate a decrease in western tropical Indian Ocean (WTIO) variability from the pre-1999 (blue bars) to the post-1999 (red bars), with the exception of 14 models generating an increase (indicated by gray shading). The multi-model mean decrease of 10% is statistically significant above the 95% confidence level based on the Bootstrap test (see "Bootstrap test" in "Methods" section). **b** A total of 37 out of the 45 models (82%) generate a decrease in WTIO variability from the present-day (1900–1999; blue bars) to the future

climate (2000–2099; red bars), with the exception of 8 models generating an increase (indicated by gray shading). The multi-model mean decrease of 13% is statistically significant above the 95% confidence level based on the Bootstrap test. Models with available geopotential height are indicated in purple. The weakened WTIO sea surface temperature (SST) variability is consistent with projected WTIO change under greenhouse warming, suggesting the reduced impact is likely to continue. Source data are provided as a Source Data file.

## Maximum covariance analysis

Maximum covariance analysis (MCA) is a singular value decomposition to get the paired spatial and temporal patterns in order[51]. The first mode of MCA maximizes the covariance and obtains the dominant coupled modes between tropical Indian Ocean SST and Antarctic SIC. Based on the Monte Carlo test, the confidence level is given by repeating 100 times MCA with the original Antarctic SIC anomalies and randomly scrambled tropical Indian Ocean SST anomalies[52].

In this study, we display MCA results in ASO, when there is highest and significant squard covariance, and establish the mechanistic link between these two modes. That is, the WTIO SST anomalies excite anomalous atmospheric Rossby wave trains, modulating Antarctic SIC anomalies reinforced by storm tracks. In other seasons with no significant squared covariance, such as November-December-January, the relationship between the WTIO and Antarctic SIC is weak, and the teleconnection is vastly different because of the unsuitable subtropical Rossby waveguide[53,54].

## t-test

We perform t-test for regression maps, and regions with the 95% confidence level are marked. The two-tailed t-test is used to examine the change of mean state (Supplementary Fig. 11).

## Strong positive IOD and moderate positive IOD

We apply the first two principal components (PCs) of the EOF analysis to SST anomalies in an equatorial Indian Ocean region (5° S–5° N, 40° E–100° E) for the ASO 1979–2020 period to separate and construct

the strong and moderate positive IOD. These two PCs exist in a non-linear relationship, expressed as $PC2(t) = \alpha[PC1(t)]^2 + \beta PC1(t) + \gamma$, with $\alpha$, $\beta$ and $\gamma$ are the nonlinear coefficient, linear coefficient, and constant of the quadratic function, respectively. EOF1 shows a cooling center off Sumatra–Java and warming in the western and central tropical Indian Ocean basin, analogous to positive IOD; EOF2 exhibits a cooling maximum in the eastern equatorial region[25]. For a strong positive IOD, the mutually reinforcement of positive EOF1 and positive EOF2 displays a strong eastern cold anomaly dominated pattern, as described by the S-index $((PC1 + PC2)/\sqrt{2})$. For a moderate positive IOD, negative EOF2 superimposes on positive EOF1, such that it is dominated by western-central warm anomalies and can be defined as the M-index $((PC1 - PC2)/\sqrt{2})$ (ref. 25).

## Sliding correlation and sliding standard deviation

We use sliding correlation and sliding standard deviation to several time series to get the change characteristics of them.

## Rossby wave source and wave activity flux

The Rossby wave source can be derived from the barotropic vorticity equation at 200 hPa

$$\frac{\partial \zeta_a}{\partial t} + \mathbf{v} \cdot \nabla \zeta_a = -\zeta_a D - F \qquad (1)$$

where $\zeta_a$ denotes the vertical component of absolute vorticity; $D$ is divergence and $F$ is the frictional term. The horizontal velocity field $\mathbf{v}$ can be split into rotational $\mathbf{v}_\psi$ and divergent $\mathbf{v}_\chi$ components, where

$\nabla \cdot \mathbf{v}_\chi = D$. Then the barotropic vorticity equation can be rewritten as

$$S = -\zeta_a D - \mathbf{v}_\chi \cdot \nabla \zeta_a \qquad (2)$$

where $S$ denotes the Rossby wave source[4].

Zonal and meridional components of the wave activity flux on the pressure coordinates are calculated using[30,31]

$$\mathbf{W} = \frac{1}{2|\mathbf{U}|} \times \begin{pmatrix} u\left(\psi_x'^2 - \psi'\psi_{xx}'\right) + v\left(\psi_x'\psi_y' - \psi'\psi_{xy}'\right) \\ u\left(\psi_x'\psi_y' - \psi'\psi_{xy}'\right) + v\left(\psi_y'^2 - \psi'\psi_{yy}'\right) \end{pmatrix} \qquad (3)$$

where $\mathbf{U} = (u, v)$ is the basic flow, with $u$ and $v$ representing zonal (eastward) and meridional (northward) wind velocity, respectively; $\psi$ is the quasi-geostrophic stream function. Perturbations are denoted by primes and the subscripts $x$ and $y$ are derivative in the zonal and meridional direction, respectively.

### Eddy-induced geopotential height tendency

To isolate the forcing of the storm tracks, we diagnose eddy-induced geopotential height tendency using geopotential height tendency ($Z_{tend}$) equation[34–36]

$$\frac{\partial \bar{Z}}{\partial t} = \left[\nabla^2 + f^2 \frac{\partial}{\partial p}\left(\frac{1}{\sigma}\frac{\partial}{\partial p}\right)\right]^{-1} \times \left\{-\frac{f}{g}\nabla \cdot \overline{\mathbf{V}_h'\zeta'} + \frac{f^2}{g}\frac{\partial}{\partial p}\left[\frac{\nabla \cdot \overline{\mathbf{V}_h'\theta'}}{-(\partial\Theta/\partial p)}\right]\right\} \qquad (4)$$

$Z_{tend}$ is associated with transient eddy vorticity (i.e., $-\frac{f}{g}\nabla \cdot \overline{\mathbf{V}_h'\zeta'}$, $\overline{F_{eddy}}$) and transient eddy heating forcing (i.e., $\frac{f^2}{g}\frac{\partial}{\partial p}\left[\frac{\nabla \cdot \overline{\mathbf{V}_h'\theta'}}{-(\partial\Theta/\partial p)}\right]$, $\overline{Q_{eddy}}$). Transient eddy vorticity and transient eddy heating forcing represent the convergence of vorticity flux and heat flux transport by transient eddies, respectively. Convergence (divergence) of the eddy vorticity flux corresponds to positive (negative) $\overline{F_{eddy}}$ anomaly, and $\overline{Q_{eddy}}$ is proportional to the vertical gradient of the heating within a certain region. Thus, the atmospheric transient eddy activities can influence the monthly mean atmospheric circulation by transporting both vorticity and heat fluxes. In our study, responses of $Z_{tend}$ induced by $\overline{Q_{eddy}}$ to the WTIO is slightly baroclinic, while responses of $Z_{tend}$ induced by $\overline{F_{eddy}}$ to the WTIO is barotropic (Supplementary Fig. 6). The total effect induced by $\overline{F_{eddy}}$ and $\overline{Q_{eddy}}$ is barotropic (Fig. 3d and Supplementary Fig. 5b). When the transient eddy forcing (maximum tendency ~6 m day$^{-1}$) lasts over 3 days, it is strong enough to drive the height anomaly[32].

### Meridional wind speed variance

Monthly variance of the 2–8-day band-pass filtered $\langle v'v' \rangle$ at 200 hPa represents storm track activities, where $v'$ is 2–8-day band-pass filtered meridional wind.

### Meridional heat transport and vertically integrated horizontal moisture flux

We calculate the poleward meridional heat transport use $MHT = -vT$, where $v$ is the meridional wind and $T$ is the temperature at 1000 hPa. To examine changes in water vapor over the Antarctic, the vertically integrated horizontal moisture flux is calculated as follows $Q = \int_0^{P_0} \frac{\mathbf{U}q}{g}dp$, where $q$ is the specific humidity (kg kg$^{-1}$) and $\mathbf{U}$ is the horizontal wind; $P_0 = 1000$ hPa and $g$ is the gravitational acceleration.

### CAM5 model simulations

To verify the teleconnection link and to establish the direction of the causal relationship between Indian Ocean SST forcing and Antarctic regional climate, we perform atmospheric general circulation model (AGCM) experiments using the Community Atmosphere Model version 5 (CAM5). CAM5 is the atmospheric component of the Community Earth System Model 1.2 with a global horizontal resolution of about 2° × 2°. The Community Land Model and the thermodynamic module of the Community Sea Ice Model are used to estimate the heat flux at the surface, to ensure that the SST is the only external lower-boundary forcing on the atmospheric model[55].

A control run is forced with monthly SST and SIC climatology, obtained by averaging HadISST and version 2 of the NOAA weekly optimum interpolation (OI.v2) SST analysis data over the period 1982–2001. For the forced run, SST anomalies were imposed following the WTIO SST pattern (Fig. 2c) with no amplitude scaling and linearly interpolated boundaries. The response of the atmospheric model to SST forcing was obtained by computing the difference between forced run and the control run. The model was integrated for 40 years, and the first 10 years were discarded as it takes a few years to assess the influence of natural variability and determine the importance of the simultaneous response. This CAM5 experiment contains the WTIO-generated teleconnections, with an atmospheric bridge starting from the tropical upper troposphere and linking to Antarctica through Rossby wave trains.

### Indian Ocean pacemaker experiments

To provide further evidence for the influence of the WTIO alone, we analyze a 10-member ensemble of Indian Ocean pacemaker experiments, in which time-evolving SST anomalies in the tropical Indian Ocean (15° S–15° N, the African coast to 180° W, with a linearly tapering buffer zone that extends to 20° S and 20° N) are nudged to observations (NOAA Extended Reconstruction Sea Surface Temperature version 3: ERSSTv3b) and the rest of the coupled climate system free to evolve. Each member of this ensemble starts with slightly different initial conditions. External forcing (both anthropogenic and natural) applied in this experiment is the same as that in CMIP5 historical simulations (1920–2005) and Representative Concentration Pathway version 8.5 (2006–2013)[56,57]. We use the ensemble-mean result, which could separate the effects of the WTIO, to analyze ASO atmospheric and Antarctic responses during the 1979–1999 period.

### CMIP6 model simulations

To investigate the impact of greenhouse warming on the WTIO changes, we use monthly SST outputs from 45 CMIP6 models (in which 32 modes are available for geopotential height) over the 1900–2099 period. The CMIP6 models are forced with historical anthropogenic and natural forcings until 2014, and thereafter with future greenhouse gases of the SSP5-8.5 emission scenario (ref. 58). Monthly anomalies are obtained with reference to the monthly climatology of the whole period and quadratically detrended. The influence of tropical Pacific is removed in Supplementary Fig. 15. For 32 models with available geopotential height, the intermodel consensus on the weakened WTIO is still strong (Fig. 5 and Supplementary Fig. 14). A total of 24 out of 32 models (75%) and 27 out of 32 models (84%) simulate a decreased WTIO variability between the pre-1999 and post-1999 periods and between the present-day and future, respectively. Their ensemble-mean changes are statistically significant.

### Bootstrap test

We use the Bootstrap method[59] to examine whether the multi-model mean decrease in the WTIO variance is statistically significant. Each model is resampled randomly to conduct 10,000 realizations of the multi-model mean value for 1900–1999 and 2000–2099 periods (Fig. 5). In this random resampling process, any model can be selected more than once. We calculate the standard deviation of the 10,000 realizations for each period. If the multi-model mean value difference between the two periods is greater than the sum of the two separate 10,000-realization standard deviation values, it is considered statistically significant above the 95% confidence level.

## Data availability

All data used in this study are available online or from the corresponding author on request. For observational data, the ERA5 data are available at https://www.ecmwf.int/en/forecasts/dataset/ecmwf-reanalysis-v5; HadISST at https://www.metoffice.gov.uk/hadobs/hadisst/data/download.html; OISST v2 at https://www.esrl.noaa.gov/psd/data/gridded/data.noaa.oisst.v2.html; ORA-S5 at https://icdc.cen.uni-hamburg.de/daten/reanalysis-ocean/easy-init-ocean/ecmwf-oras5.html; SODA3.12.2 at https://www2.atmos.umd.edu/~ocean/index_files/soda3.12.2_mn_download_b.htm; GODAS at https://www.esrl.noaa.gov/psd/data/gridded/data.godas.html; the Niño3.4 index at https://psl.noaa.gov/gcos_wgsp/Timeseries/Nino34; the DMI at https://psl.noaa.gov/gcos_wgsp/Timeseries/DMI/. The Indian Ocean pacemaker experiment data can be downloaded from https://www.cesm.ucar.edu/working-groups/climate/simulations/cesm1-indian-ocean-pacemaker. The CMIP6 data can be downloaded from https://esgf-node.llnl.gov/projects/cmip6/. Source data are provided with this paper.

## Code availability

The Matlab2021b is used for plotting. Codes to reproduce the figures of this study are provided as a Source Data file.

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

## Acknowledgements

This study is supported by National Natural Science Foundation of China (42376009), Strategic Priority Research Program of Chinese Academy of Sciences (XDB40000000), and National Key Research and Development Program of China (2023YFF0806700) of L.Z.

## Author contributions

L.Z. and W.C. conceived the research. L.Z. and X.R. wrote the initial manuscript. X.R. performed all analyses and generated final figures. L.Z., X.R., W.C., X.L., and L.W. contributed to interpreting results, discussing the associated dynamics, and improving this paper.

## Competing interests

The authors declare no competing interests.
