## [Peer Review File · Nature Communications]

Weakened western Indian Ocean dominance on Antarctic sea ice variability in a changing climateEditorial Note: Parts of this Peer Review File have been redacted as indicated to remove third-party material where no permission to publish could be obtained.

REVIEWER COMMENTS

Reviewer #1 (Remarks to the Author):

The manuscript shows a consistent decrease in the impact of WTIO SST on Antarctic sea ice. While the cause is not explicitly identified from observations alone, climate model simulations suggest a link to greenhouse warming, indicating a reduction in WTIO SST variability. The paper emphasizes the robustness of the findings, supported by multiple analyses and comparisons between observed and simulated data. The paper concludes by suggesting that the projected decrease in inter-annual influence from the Indian Ocean could contribute to an accelerated decline in Antarctic sea ice.

This article is suitable to be published because of the different approaches used in the analysis and also emphasizes the robustness of the findings, supported by multiple analyses and comparisons between observed and simulated data. In addition, the paper challenges previous assumptions about the primary drivers of Antarctic sea ice variability. However, some improvements are needed before publishing. Also, there are many occasions where the explanations are unclear and confusing, and a thorough reading is necessary to avoid the errors. Refer to the manuscript for further comments.

Reviewer #2 (Remarks to the Author):

Remarks: Authors discussed the role of the Indian ocean to the Antarctic sea ice variability using observational, model datasets and model experiments. I have gone through manuscript, and I think the methods and results are not sufficient enough to explain the presented hypothesis. Please see some of the major comments below. Unfortunately, I do not recommend the manuscript in its present form for the journal publication.

1- For model experiments, authors consider the AGCM simulations, which do not fits with the presented hypothesis, where authors explain the change in the WTIO/sea-ice relationship is due to the changes in Indian Ocean (IO) changes, mainly western IO. To explain this, Ocean and ocean-atmosphere dynamics is important to be considered to discuss such changes related the tropical basin mean or variability. This is one of the main discrepancies of this article. Also, for experiment, is Co2 fixed?

2- Extended Data4: It looks more like a decadal variability, which is also showing a sign of recovering the relationship around 2007/08? I do not think it's a weakening in the relationship. It looks more like a decadal variability! I also suggest authors to consider a long time series. For example, ERA5 data is available for the long term and that could be useful.

3- Fig. 4: This is for CMIP6 models, which are coupled models. Now the period is about 100 years, where they compared the baseline with the future period, which is fine. However, going with point (1), authors need to re-consider the modelling strategy.

4- Authors discussed about the different IOD flavours? But how much their impacts are distinguishable to sea-ice variability is not clear.

5- Mechanism: lne 151-152: Authors talked about tropical convection. Are they considering the lag response of SSTs to the tropical convection? or is this in the early austral summer season? If it's a lead/lag response, it's really important to explain, how the SST signal influences to the tropical convection? And to explain this, coupled model simulation is more appropriate, than AGCM.

Reviewer #3 (Remarks to the Author):

General comments

This manuscript provides an interesting contribution on how the IOD is changing and how that is affecting its teleconnections and impacts upon Antarctic sea ice. The work deserves publication, but the text needs quite a bit of revision. I found it very dense and hard to follow in a lot of places. Try to spell out the key findings clearly, that the IOD affects sea ice via a Rossby wave teleconnection, that the form of that teleconnection is changing/weakening over time and this impacts the teleconnection to Antarctic sea ice, and the observed trends are likely to continue in the future, based on model simulations. Aim to keep the statistical discussion constrained. Also, reflect more on the robustness of these results, given the short time series you're worki

ng with. I wonder about the efficacy of the "ENSO signal removal" for this work as the IOD and ENSO are quite strongly associated. I appreciate the statistical findings are backed up by model simulations and you have applied good significance testing, but some of the results hang on subtle changes in regression patterns that are bound to be sensitive to sampling, in my opinion.

The English style is generally good, although the text is quite hard to read and follow in many places. There are grammatical slips in places (some of which are identified below). Please have the manuscript carefully proof-read by a native English speaker, and preferably a technical editor, before final submission.

Specific comments

1. Lines 33-66: Somewhere in this introductory page it would be good to mention that all the data sets analysed have had an ENSO signal linearly removed by regression. This is discussed in the Methods section later but it would be useful to alert the reader to this before showing any results.
2. Lines 38-39: Reword "...anomalous cooling in the east drives anomalous sinking motion over the eastern tropical, and anomalous warming in the west leads to anomalous ascending." to ""...anomalous cooling in the east drives anomalous sinking motion over the eastern tropical Indian Ocean, and anomalous warming in the west leads to anomalous ascending motion there."
3. Line 41: Reword "...forcing nonlocal Rossby wave source" to "...generating a Rossby wave source"
4. Line 42: Reword "...wave trains, curving..." to "...wave trains curve..."
5. Lines 69-85: The result discussed here, that of Fig 1, and the associated Extended Fig 1, are very interesting and seem to make a coherent story. I take it that Fig 1 results come from data that have had

an ENSO signal linearly regressed out while in Extended Fig 1 the ENSO signal is still there. I wonder if you have looked at panels analogous to (e) and (f) but using the ETIO time series? If you did and the result was much less of a match with the MCA results, that would strengthen your result further. You talk about this later which does close the loop I guess, but I find the sequencing of the discussion hard to follow.

6. Lines 86-102: I had to read this paragraph several times, with reference to Fig 1 and Extended Fig 2. I get the sense of what you're saying but it is not written as clearly as it could be. The statements on lines 98-100 about correlations between WTIO & ETIO SSTs and the M & S time series of the IOD is not informed by Extended Fig. 2, as far as I can see. The spatial patterns shown support the sense of what is said but the correlations quoted are not illustrated in the Figure. Lines 100-102 summarise the relevant relationships well, with the quoted correlation coefficients.

7. Lines 103-123: Some of the differences in regression patterns discussed are quite subtle. As you say, IOD events tend to co-occur with ENSO events. I wonder about the physical meaning of the ENSO-removed data and results as I don't think one can really separate the two phenomena this way. Again, I find the discussion difficult to tease out, and conclusions seem to rest on small differences between regression patterns. How sensitive are these results to the sample size? If you added the last couple of years, does that change anything much?

8. Lines 126-146: This section is easier to follow, and Fig. 2 is informative. One wonders though about the meaning or robustness of the signal, given the relatively short period of time being analysed.

9. Lines 195-196: Where you say the SST anomalies in the model runs are "imposed following the WTIO SST pattern", do you mean using the spatial patterns as shown in Figs 1 or 2, at the shown amplitude? At this point I am not clear what the WTIO SST pattern is, exactly. Is there any amplitude scaling?

10. Line 214: Say "...over which reliable observations are available."

11. Line 271: The final statement that "melt of Antarctic sea ice could accelerate", based on changes in the IOD teleconnection, seems unsupported to me. Yes, changes in the phenomena you identify will affect the nature of interannual variability in sea ice extent, but why would they encourage faster ice melt?

We thank the three reviewers for their helpful comments. Specifically, we have tried our best to enhance clarity throughout the manuscript, as suggested by Reviewers #1 and #3. We have used fully coupled models and observational datasets to test the robustness of the results, as suggested by Reviewer #2, and we are delighted to report that our original result is strengthened.

Our responses (in blue) to reviewer's comments (in black) are as follows. All page and figure numbers pertain to the revised version, i.e., "Revised_Manuscript.docx" file. And the detailed revision track is in the file "Revised_Manuscript_with_track.docx".

Reviewer #1 (Remarks to the Author):

The manuscript shows a consistent decrease in the impact of WTIO SST on Antarctic sea ice. While the cause is not explicitly identified from observations alone, climate model simulations suggest a link to greenhouse warming, indicating a reduction in WTIO SST variability. The paper emphasizes the robustness of the findings, supported by multiple analyses and comparisons between observed and simulated data. The paper concludes by suggesting that the projected decrease in inter-annual influence from the Indian Ocean could contribute to an accelerated decline in Antarctic sea ice. This article is suitable to be published because of the different approaches used in the analysis and also emphasizes the robustness of the findings, supported by multiple analyses and comparisons between observed and simulated data. In addition, the paper challenges previous assumptions about the primary drivers of Antarctic sea ice variability. However, some improvements are needed before publishing. Also, there are many occasions where the explanations are unclear and confusing, and a thorough reading is necessary to avoid the errors. Refer to the manuscript for further comments.

Thank you for your comment. We have revised our manuscript following your comment in the annotated manuscript in the "Decision" file. Below we provide clarification on unclear/confusing text.

1. Lines 45-46. Not clear: "...Australia rainfall and enhanced bushfires.":

During positive IOD events, the associated teleconnection is conducted via an anomalous atmospheric Rossby wave train, with a high-pressure anomaly around southern Australia, which reduces rainfall there. This promotes anomalous drought, together with suitable windy and hot weather conditions, raising bushfire risks. In East African countries, floods and malaria outbreaks, occur.

2. Line 66. It is better to add a schematic diagram as it is difficult to visualize.

Thanks for this suggestion, and we add a schematic diagram, **Supplementary Fig. 16** (see below), to better visualize.

Supplementary Fig. 16 | Dominant atmospheric teleconnection associated with western Indian Ocean influence on Antarctic sea ice variability. Highlighted in the schematic are responses in the surface and upper troposphere. Warm WTIO SST anomalies (red shading) lead to changes in tropical convection (cloud with white arrows) driving anomalous divergent flow (gray arrows) in the upper troposphere. The divergent flow, in turn, excites Rossby wave sources (red and light blue vortices) in the subtropics and forces Rossby wave trains (pink lines) curve poleward and eastward towards Antarctica with alternating centers of high- and low-pressure anomalies (red and blue circles). The barotropic atmospheric response in SLP (green and orange shadings) alters Antarctic sea ice (red and light blue bold lines) through thermal advection and wind-driven drift (red and blue arrows). The pattern is distinctively different from that ENSO-induced (gray dashed line), and is reinforced by a synoptic-eddy feedback {Jet stream (orange arrow) with storm tracks (red helixes)}. This WTIO dominance of the Indo-Pacific influence on Antarctic sea ice has been weakening in line with projected change in the IOD by climate models (gradient orange shading).

3. Lines 92-93. I'm uncertain about the meaning of this: "The WTIO time series by and large, represents the MCA-SST time series ($r = 0.94$), but not the ETIO time series as reflected by their weak correlation ($r = 0.23$)":

We have revised our manuscript as "As expected, the time series of WTIO ($r = 0.94$), rather than that of ETIO ($r = 0.23$), largely represents the first MCA SST time series, whose corresponding SST pattern is dominated by the WTIO.", in **L99–101**.

4. Lines 98-102. I'm uncertain about the meaning of this: “The WTIO and ETIO SST time-series show strong correlations with the moderate IOD and strong IOD indices, at 0.71 and 0.91, respectively (Supplementary Fig. 2). However, the DMI primarily reflects the ETIO and strong IOD variability ($r = 0.84$ and 0.86 , respectively), whereas the relationship of the DMI with WTIO or with moderate IOD is much weaker ($r = 0.22$ or 0.32 , respectively)”:

We have revised our manuscript as “The DMI primarily represents ETIO-dominated IOD events, as reflected in its strong correlation with the ETIO ($r = 0.84$) or strong IOD ($r = 0.86$). However, the relationship of the DMI with the WTIO ($r = 0.22$) or with moderate IOD ($r = 0.32$) is much weaker. Thus, DMI might not be effective for identifying the impact from the WTIO.”, in **L108–112**.

5. Lines 120-123. Yellow shading: “Here, we find that positive WTIO SST variability dominates the Indo-Pacific’s influence on the principal pattern of austral spring Antarctic SIC variability, which features a decrease in sea ice near the Amundsen Sea, but an increase to regions both sides as shown in Fig. 1f”:

This is a short summary of our finding, and we have revised; it now reads as “Thus, WTIO SST variability predominantly affects the principal pattern of austral spring Antarctic SIC variability, which at a positive phase features a decrease in sea ice near the Amundsen Sea, but an increase in regions both sides.”, in **L119–121**.

Reviewer #2 (Remarks to the Author):

Remarks: Authors discussed the role of the Indian ocean to the Antarctic sea ice variability using observational, model datasets and model experiments. I have gone through manuscript, and I think the methods and results are not sufficient enough to explain the presented hypothesis. Please see some of the major comments below. Unfortunately, I do not recommend the manuscript in its present form for the journal publication.

1- For model experiments, authors consider the AGCM simulations, which do not fits with the presented hypothesis, where authors explain the change in the WTIO/sea-ice relationship is due to the changes in Indian Ocean (IO) changes, mainly western IO. To explain this, Ocean and ocean-atmosphere dynamics is important to be considered to discuss such changes related the tropical basin mean or variability. This is one of the main discrepancies of this article. Also, for experiment, is Co2 fixed?

Thank you for this comment. We consider the reply from the following two perspectives.

Firstly, the tropical SST forcing can drive teleconnection from the tropics to the Antarctic, with an atmospheric bridge starting from the tropical upper troposphere and to the Antarctica through Rossby wave trains. To verify such teleconnection link and establish the direct causal link between the WTIO SST anomalies and the Antarctic atmospheric pattern, we perform the AGCM experiment. If the AGCM cannot establish the teleconnection, even if similar teleconnection exists in coupled experiments, we would not be able to pinpoint the source of SST variability that dominates the tropical Indo-Pacific impact on Antarctic sea ice.

Following your suggestion, we analyse Indian Ocean pacemaker experiments, conducted in a fully coupled climate model.

This Indian Ocean pacemaker experiments contain a ten-member ensemble of Community Earth System Model1 simulations, in which time-evolving SSTAs in the tropical Indian Ocean (15°S–15°N, the African coast to 180°W, with a linearly tapering buffer zone that extends to 20°S and 20°N) are nudged to observations (NOAA Extended Reconstruction Sea Surface Temperature version 3: ERSSTv3b) and the rest of the model's coupled climate system free to evolve (Fig. r1-1). External forcing (both anthropogenic and natural) applied in this experiment is the same as that in CMIP5 historical simulations (1920–2005) and Representative Concentration Pathway version 8.5 (2006–2013) (refs. 1,2). The nudging region in this set of experiments includes the entire tropical Indian Ocean as well as the IOD diversity. Thus, the ensemble mean of Indian Ocean pacemaker experiments could separate roles of the WTIO SST in the extratropical responses.

The results of the pacemaker experiments (Fig. r1-2c and r1-2d; i.e., Fig. 4c and 4d) are

similar to results from the reanalysis (Fig. 3b and 3d) and the AGCM experiments (Fig. r1-2a and r1-2b; i.e., Fig. 4a and 4b). Associated with the WTIO warm SSTA pattern, anomalous Rossby wave train pattern shows Z200 and SLP alternative high and low anomaly centers in the mid- and high-latitudes of the Southern Hemisphere (shading in Fig. r1-2c and r1-2d; i.e., Fig. 4c and 4d), which affect regional-scale SAT and SIC anomalies around the Antarctic (contours and left bottom corner in Fig. r1-2d; i.e., Fig. 4d).

Accordingly, WTIO SST-induced teleconnection can be seen in a) reanalysis, b) the AGCM, and c) the Indian Ocean pacemaker experiment results. The consistency from three independent analyses confirms our finding.

We have revised our manuscript (L225–237) and figure (Fig. 4).

Secondly, in the AGCM experiment, the CO₂ is fixed. As previously mentioned, AGCM is used to identify the source of SST variability and to establish the mechanistic atmospheric bridge. That is why CO₂ is fixed.

In the CMIP6 experiments, we consider the changes in CO₂ as an important factor influencing WTIO SST variability (L253–257), representing the impact of greenhouse warming on the WTIO, and consequently on the WTIO-related teleconnection.

[REDACTED]

Figure r1-1. | Forcing region. Shading represents the nudging region in the Indian Ocean pacemaker experiments. SSTAs (15°S–15°N, African Coast to 174°E, with a linearly tapering buffer zone that extends to 20°S and 20°N) nudged to ERSSTv3b observations^{1,2}. This figure comes from the website: <https://www.cesm.ucar.edu/working-groups/climate/simulations/cesm1-indian-ocean-pacemaker>.

Figure r1-2 (i.e., Fig. 4) | Numerical model experiments assessing WTIO SST anomalies effects. a, b Simulated a Z200 (m), b SLP (shading; Pa), and SAT (contours; positive magenta and negative green; zero line omitted; starts from ± 0.3 °C and interval ± 0.25 °C) responses to WTIO SST forcing (i.e., Fig. 2c, pre-1999). **See the ‘CAM5 model simulations’ section in the Methods.** c, d Regressions of c Z200 (m), d SLP (shading; Pa), SAT (contours; positive magenta and negative green; zero line omitted; starts from ± 0.3 °C and interval ± 0.3 °C), and SIC (left bottom corner; positive purple and negative green) onto normalized WTIO time series in the Indian Ocean pacemaker experiments for the pre-1999 period. **See the ‘Indian Ocean pacemaker experiments’ section in the Methods.** Stippling indicates the 95% confidence level based on the *t*-test. The CAM5 simulations and Indian Ocean pacemaker experiments display a consistent atmospheric bridge, which is similar to the reanalysis result.

2- Supplementary4: It looks more like a decadal variability, which is also showing a sign of recovering the relationship around 2007/08? I do not think it’s a weakening in the relationship. It looks more like a decadal variability! I also suggest authors to consider a long time series. For example, ERA5 data is available for the long term and that could be useful.

Thank you for this comment. We consider your comments from the following two perspectives.

(1) Observational studies on sea ice are based on the period since satellite observations in 1979. Although ERA5 data are available for the long term (i.e., from 1940 to the present), the SIC data are unreliable for the 1940–1978 periods³. Before 1979, the HadISST2 dataset was used. Specifically, prior to 1973, the SIC analysis was derived indirectly and did not include any interannual variability; for 1973–1978, the SIC was digitized by the U.S. National Climatic Data Center (NCDC) from hand-drawn U.S. National Ice Center (NIC) analyses. Thus, it gives only a general indication of sea ice extent variations in the Southern Hemisphere on decadal timescales before the 1970s³. Fig. r2-1 shows the SIC EOF PC1 from 1940 to the present, indicating that the issues existed before 1979.

Figure r2-1 | Antarctic SIC EOF PC1 for the 1940–2020 ASO period.

Thus, the sliding correlation between the WTIO time series and SIC EOF PC1 is unachievable in the long term. But we can examine WTIO SST variability.

Antarctic SIC anomalies, acting as a “dependent variable”, are influenced by WTIO SST variability. We use long term ERA5 SST data and add five other SST reanalysis products to examine our finding of the weakening of WTIO SST variability. These six products are:

- ERA5 (extends to long term, from 1940 to 2022) (ref⁴);
- HadISST (Hadley Centre Sea Ice and Sea Surface Temperature data set from 1940 to 2021) (ref³);
- OISST v2 (NOAA Optimum Interpolation SST version 2 from 1982 to 2021) (ref⁵);
- ORA-S5 (ECMWF Ocean Reanalysis System 5 from 1958 to 2022) (ref⁶);
- SODA3.12.2 (Simple Ocean Data Assimilation version 3.12.2 from 1980 to 2017) (ref⁷);
- GODAS (NCEP Global Ocean Data Assimilation System from 1980 to 2017) (ref⁸).

These six SST data sets show consistent WTIO 21-year sliding standard deviation results (Fig. r2-2). Focusing on the period after 1978, all of these six data sets reproduce the pronounced and coherent reduction of WTIO SST variability in recent decades.

We have revised our manuscript (L133–137) and figure (Fig. 2).

Figure r2-2 A combination of six reanalysis SST products consistently shows a reduction of WTIO SST variability around 1999. The x-axis indicates the starting year in the sliding window.

(2) To distinguish the impact of global warming and decadal variability, it is necessary to use the CMIP6 multi-model mean result. Each CMIP6 model simulates independent decadal climate variability, with evolution differs vastly from one model to another. The multi-model mean result of CMIP6 reflects the influence of climate change (greenhouse warming) because the independent variability would be removed in the ensemble average.

The multi-model mean of the 21-year sliding standard deviation (i.e., models without grey shading in Fig. 5a) suggests a decrease in WTIO variability since the 1980s (Fig. r2-3; i.e., Supplementary Fig. 12). This reduction is consistent with the combination of six reanalysis products over the observation period (Fig. r2-2). The rates of decrease in WTIO SST variability in multi-model and observational data are -0.093 standard deviation decade⁻¹ and -0.208 standard deviation decade⁻¹, respectively, indicating that greenhouse warming likely contributes to the observed decrease by 45% (-0.093 divided by -0.208).

We have revised our manuscript (L266–270) and figure (Supplementary Fig. 12).

Figure r2-3 (i.e., Supplementary Fig. 12) | 21-year sliding standard deviation of WTIO time series in CMIP6. Solid blue line and shading indicate selected multi-model mean

(i.e., a total of 31 out of the 45 models without gray shading in Fig. 5a) and 1.0 standard deviation of a total of 10,000 inter-realizations based on a bootstrap method for the 1979–2020 period, respectively. The x-axis indicates the starting year in the sliding window. The declining trend is -0.093 standard deviation decade⁻¹.

3- Fig. 4: This is for CMIP6 models, which are coupled models. Now the period is about 100 years, where they compared the baseline with the future period, which is fine. However, going with point (1), authors need to re-consider the modelling strategy.

Thank you for this comment.

As mentioned in response to your Comment 1, we identify WTIO-dominated teleconnection by three independent methods including a) reanalysis, b) AGCM simulations, and c) Indian Ocean pacemaker experiments.

4- Authors discussed about the different IOD flavours? But how much their impacts are distinguishable to sea-ice variability is not clear.

The ETIO-induced SIC anomaly is small and statistically insignificant, although the ETIO SSTA amplitude is larger than that of WTIO.

To compare the WTIO and ETIO impacts on SIC, we calculate the sum of SIC anomaly amplitude for each grid in these two cases (i.e., $\sqrt{\sum_{i=1}^k SIC_i^2}$, where k is the number of grids with absolute SIC anomaly $> 1.6\%$ spanning $50^{\circ}\text{S}–90^{\circ}\text{S}$, $120^{\circ}\text{E}–180^{\circ}–30^{\circ}\text{W}$; patterns in Fig. r4-1; i.e., Supplementary Fig. 3). The values associated with the WTIO and the ETIO are 42 and 21, respectively, which means that the impact of the WTIO on SIC is about two times that of the ETIO on SIC.

We have revised our manuscript (L113–121) and figure (Supplementary Fig. 3).

Figure r4-1 (i.e., Supplementary Fig. 3) | Response of Antarctic SIC to ETIO SST anomalies and DMI. a, b Regressions of **a** SST (°C) and **b** SIC anomalies (%) onto normalized WTIO time series for the ASO 1979–2020 period. **c, d** Same as **a, b**, but onto normalized ETIO time series. **e, f** Same as **a, b**, but onto normalized DMI. Stippling indicates the 95% confidence level based on the *t*-test.

5- Mechanism: lne 151-152: Authors talked about tropical convection. Are they considering the lag response of SSTs to the tropical convection? or is this in the early austral summer season? If it's a lead/lag response, it's really important to explain, how the SST signal influences to the tropical convection? And to explain this, coupled model simulation is more appropriate, than AGCM.

There is no evidence to support a substantial lead/lag response.

(1) In the tropics, the interaction between the ocean and the atmosphere is asymmetric, predominantly characterized by the ocean forcing the atmosphere. This process is completed within a few days, and the atmospheric signals cannot persist for more than two months. For atmospheric teleconnection, Rossby wave trains generated by tropical SSTAs reach around the Antarctic^{9,10,11} in less than two weeks. Therefore, it is reasonable to focus on the simultaneous response of the Antarctic to

the tropics.

We have revised our manuscript (L154).

(2) According to the Rossby wave theory, there are unsuitable background conditions in early austral summer (November-December; ND; or November-December-January; NDJ) for Rossby wave to propagate. This is because the subtropical jet is too weak to counteract the planetary vorticity gradient and unable to keep Rossby wave activity trapped in the Southern Hemisphere^{10,11} (Fig. r5-1).

This is also why we see the squared covariance of the first MCA mode (Fig. 1a) reaching a peak in ASO and then rapidly declining, and becoming insignificant during the summer.

Figure r5-1. 200 hPa zonal wind (upper panel) and stationary wave number (Ks; lower panel) for the 1979–2020 ASO (left panel) and ND (right panel) periods. Subtropical and polar white regions in Ks denote the wave reflecting surface, keeping the Rossby wave activity trapped in the extratropic except austral summer.

References:

1. Yang, D. et al. Role of tropical variability in driving decadal shifts in the Southern Hemisphere summertime eddy-driven jet. *J. Clim.* (2020).
2. Zhang, L., Han, W., Karlsruhas, K. B., Meehl, G. A., Hu, A., Rosenbloom, N. & Shinoda T. Indian Ocean Warming Trend Reduces Pacific Warming Response to Anthropogenic Greenhouse Gases: An Interbasin Thermostat Mechanism. *Geophys. Res. Lett.* **46**, 10882-10890 (2019).
3. Rayner, N. A. et al. Global analyses of sea surface temperature, sea ice, and night marine air temperature since the late Nineteenth Century. *Journal of Geophysical Research. J. Geophys. Res.* **108**, 4407 (2003).
4. Hersbach, H. et al. The ERA5 global reanalysis. *Q. J. R. Meteorol. Soc.* **146**, 1999–2049 (2020).
5. Reynolds, R. W., Rayner, N. A., Smith, T. M., Stokes, D. C. & Wang, W. An improved in situ and satellite SST analysis for climate. *J. Clim.* **15**, 1609–1625 (2002).
6. Zuo, H., Balmaseda, M. A., Tietsche, S., Mogensen, K. & Mayer, M. The ECMWF operational ensemble reanalysis–analysis system for ocean and sea ice: a description of the system and assessment. *Ocean Sci.* **15**, 779–808 (2019).
7. Carton, J. A., Chepurin, G. A. & Chen, L. SODA3: A new ocean climate reanalysis. *J. Clim.* **31**, 6967–6983 (2018).
8. Behringer, D. & Xue, Y. Evaluation of the global ocean data assimilation system at NCEP: the Pacific Ocean. In Eighth Symposium on Integrated Observing and Assimilation Systems for Atmosphere, Oceans, and Land Surface (American Meteorological Society, 2004).
9. Li, X., Holland, D. M., Gerber, E. P. & Yoo, C. Impacts of the north and tropical Atlantic Ocean on the Antarctic Peninsula and sea ice. *Nature* **505**, 538–542 (2014).
10. Li, X., Holland, D. M., Gerber, E. P. & Yoo, C. Rossby Waves Mediate Impacts of Tropical Oceans on West Antarctic Atmospheric Circulation in Austral Winter. *J. Clim.* **28**, 8151–8164 (2015).
11. Li, X., Gerber, E. P., Holland, D. M. & Yoo, C. A Rossby Wave Bridge from the Tropical Atlantic to West Antarctica. *J. Clim.* **28**, 2256–2273 (2015).

Reviewer #3 (Remarks to the Author):

General comments

This manuscript provides an interesting contribution on how the IOD is changing and how that is affecting its teleconnections and impacts upon Antarctic sea ice. The work deserves publication, but the text needs quite a bit of revision. I found it very dense and hard to follow in a lot of places. Try to spell out the key findings clearly, that the IOD affects sea ice via a Rossby wave teleconnection, that the form of that teleconnection is changing/weakening over time and this impacts the teleconnection to Antarctic sea ice, and the observed trends are likely to continue in the future, based on model simulations. Aim to keep the statistical discussion constrained. Also, reflect more on the robustness of these results, given the short time series you're working with. I wonder about the efficacy of the "ENSO signal removal" for this work as the IOD and ENSO are quite strongly associated. I appreciate the statistical findings are backed up by model simulations and you have applied good significance testing, but some of the results hang on subtle changes in regression patterns that are bound to be sensitive to sampling, in my opinion.

The English style is generally good, although the text is quite hard to read and follow in many places. There are grammatical slips in places (some of which are identified below). Please have the manuscript carefully proof-read by a native English speaker, and preferably a technical editor, before final submission.

Thank you for your helpful suggestions and comments.

Specific comments

1. Lines 33-66: Somewhere in this introductory page it would be good to mention that all the data sets analysed have had an ENSO signal linearly removed by regression. This is discussed in the Methods section later but it would be useful to alert the reader to this before showing any results.

Thank you for this suggestion, and we have revised in our manuscript (L51–53 and L77–80).

2. Lines 38-39: Reword "...anomalous cooling in the east drives anomalous sinking motion over the eastern tropical, and anomalous warming in the west leads to anomalous ascending." to "...anomalous cooling in the east drives anomalous sinking motion over the eastern tropical Indian Ocean, and anomalous warming in the west leads to anomalous ascending motion there."

Thank you for this suggestion, and we have revised, as suggested.

3. Line 41: Reword "...forcing nonlocal Rossby wave source" to "...generating a Rossby wave source"

Thank you for this suggestion, and we have revised in accordingly.

4. Line 42: Reword "...wave trains, curving..." to "...wave trains curve..."

Thank you for noting this, and we have revised.

5. Lines 69-85: The result discussed here, that of Fig 1, and the associated Extended Fig 1, are very interesting and seem to make a coherent story. I take it that Fig 1 results come from data that have had an ENSO signal linearly regressed out while in Extended Fig 1 the ENSO signal is still there. I wonder if you have looked at panels analogous to (e) and (f) but using the ETIO time series? If you did and the result was much less of a match with the MCA results, that would strengthen your result further. You talk about this later which does close the loop I guess, but I find the sequencing of the discussion hard to follow.

Thank you for this comment. We have revised the relevant texts (L93–121) and figure (Supplementary Fig. 3).

According to the logical chain, up until L92, the impacts of WTIO and ETIO are still not known or distinguished. The results from MCA reveal that the anomalous SST mode is predominantly driven by WTIO and is closely linked with SIC EOF1. We then point out the limitations of using DMI and discuss the need to distinguish between the WTIO and ETIO.

Anomalies of SST and SIC related to the ETIO show little match with the MCA results, with a ETIO cooling-dominated SSTA pattern and a weak, statistically insignificant SIC anomalies pattern (Fig. r5-1; i.e., Supplementary Fig. 3).

We have displayed anomalous Rossby wave train pattern associated with the ETIO in Supplementary Fig. 8c and 8d. Here, we reorganize the logical structure of this section to bring out our result (L186–193).

Figure r5-1 (i.e., Supplementary Fig. 3) | Response of Antarctic SIC to ETIO SST anomalies and DMI. **a, b** Regressions of **a** SST (°C) and **b** SIC anomalies (%) onto normalized WTIO time series for the ASO 1979–2020 period. **c, d** Same as **a, b**, but onto normalized ETIO time series. **e, f** Same as **a, b**, but onto normalized DMI. Stippling indicates the 95% confidence level based on the *t*-test.

6. Lines 86-102: I had to read this paragraph several times, with reference to Fig 1 and Extended Fig 2. I get the sense of what you're saying but it is not written as clearly as it could be. The statements on lines 98-100 about correlations between WTIO & ETIO SSTs and the M & S time series of the IOD is not informed by Extended Fig. 2, as far as I can see. The spatial patterns shown support the sense of what is said but the correlations quoted are not illustrated in the Figure. Lines 100-102 summarise the relevant relationships well, with the quoted correlation coefficients.

Thank you for this suggestion, and we have revised accordingly (L102–108).

7. Lines 103-123: Some of the differences in regression patterns discussed are quite subtle. As you say, IOD events tend to co-occur with ENSO events. I wonder about the physical meaning of the ENSO-removed data and results as I don't think one can really

separate the two phenomena this way. Again, I find the discussion difficult to tease out, and conclusions seem to rest on small differences between regression patterns. How sensitive are these results to the sample size?

Thank you for this comment. This method of separation is designed to isolate the impacts of the WTIO and ENSO on teleconnections in ASO, rather than to separate the connections between ENSO and the WTIO. In our original manuscript, we performed WTIO-forcing AGCM experiments, the result of which strongly support the regression results related to the WTIO.

Here, we add three independent analyses to strengthen our results.

Firstly, we employ a composite analysis to confirm our ENSO-removed result. We select four years, including 1979, 1992, 2003, and 2019, with an absolute value of normalized WTIO time series > 0.8 and Niño 3.4 index < 0.4 , representing WTIO type IOD events but without ENSO events (Fig. r7-1, left). The composite SST and SLP results indicate similar spatial patterns to those from regression results, although the amplitude is higher and the centers of anomalies are slightly shifted (Fig. r7-1, right).

Figure r7-1. Four WTIO dominated IOD without ENSO are selected (left; choose years with absolute value of normalized WTIO time series > 0.8 and Niño 3.4 index < 0.4 : 1979, 1992, 2003, and 2019) and the associated composite SST (right upper panel; °C) and SLP (right lower panel; Pa) anomalies.

Secondly, ENSO teleconnection is vastly different from that induced by the WTIO SST. The ENSO teleconnection in the Southern Hemisphere features the Pacific–South American (PSA) wave train, starting in the central South Pacific and deepening the Amundsen Sea Low (ASL) that alters Antarctic dipole structure SIC anomalies (Fig. r7-2; i.e., Supplementary Fig. 9). We have revised our manuscript (L199–205) and figures (Supplementary Figs. 9 and 10).

Figure r7-2. (i.e., Supplementary Fig. 9) | ENSO related teleconnection. a-e Regressions of **a** SST ($^{\circ}\text{C}$), **b** convective precipitation (mm; 15°S – 30°N), RWS (10^{-8} s^{-2} ; 45°S – 15°S), **c** Z200 (shading; m), WAF (vectors; $\text{m}^2 \text{ s}^{-2}$), **d** SAT ($^{\circ}\text{C}$), and **e** SIC (%) onto normalized Niño 3.4 index for the ASO 1979–2020 period. Stippling indicates the 95% confidence level based on the *t*-test.

Furthermore, we show consistent results from a set of Indian Ocean pacemaker experiments. The Indian Ocean pacemaker experiments contain a 10-member ensemble of Community Earth System Model1 simulations, in which time-evolving SSTAs in the tropical Indian Ocean (15°S – 15°N , the African coast to 180°W , with a linearly tapering buffer zone that extends to 20°S and 20°N) are nudged to observations (NOAA Extended Reconstruction Sea Surface Temperature version 3: ERSSTv3b) and the rest of the model’s coupled climate system free to evolve (Fig. r7-3). External forcing (both anthropogenic and natural) applied in this experiment is the same as that in CMIP5 historical simulations (1920–2005) and Representative Concentration Pathway version 8.5^{1,2} (2006–2013).

The results of the pacemaker experiments (Fig. r7-4c and r7-4d; i.e., Fig. 4c and 4d) are similar to those from the reanalysis (Fig. 3b and 3d) and the AGCM experiments (Fig. r7-4a and r7-4b; i.e., Fig. 4a and 4b). Associated with the WTIO warm SSTA pattern, anomalous Rossby wave train pattern shows Z200 and SLP alternative high and low anomaly centers in the mid- and high-latitudes of the Southern Hemisphere (shading in Fig. r7-4c and r7-4d; i.e., Fig. 4c and 4d), which affect regional-scale SAT and SIC anomalies around the Antarctic (contours and left bottom corner in Fig. r7-4d; i.e., Fig. 4d).

We have revised our manuscript (L225–237) and figure (Fig. 4).

In short, these three independent methods including a) composite, b) ENSO teleconnection analysis, c) Indian Ocean pacemaker experiments all support our results.

[REDACTED]

Figure r7-3. | Nudging region. Shading represents the nudging region in the Indian Ocean pacemaker experiments. SSTAs (15°S–15°N, African Coast to 174°E, with a linearly tapering buffer zone that extends to 20°S and 20°N) nudged to ERSSTv3b observations^{1,2}. This figure comes from the website: <https://www.cesm.ucar.edu/working-groups/climate/simulations/cesm1-indian-ocean-pacemaker>.

Figure r7-4 (i.e., Fig. 4) | Numerical model experiments assessing WTIO SST anomalies effects. a, b Simulated **a** Z200 (m), **b** SLP (shading; Pa), and SAT (contours; positive magenta and negative green; zero line omitted; starts from ± 0.3 °C and interval ± 0.25 °C) responses to WTIO SST forcing (i.e., Fig. 2c, pre-1999). **See the ‘CAM5 model simulations’ section in the Methods.** **c, d** Regressions of **c** Z200 (m), **d** SLP (shading; Pa), SAT (contours; positive magenta and negative green; zero line

omitted; starts from ± 0.3 °C and interval ± 0.3 °C), and SIC (left bottom corner; positive purple and negative green) onto a normalized WTIO time series in the Indian Ocean pacemaker experiments for the pre-1999 period. **See the ‘Indian Ocean pacemaker experiments’ section in the Methods.** Stippling indicates the 95% confidence level based on the t -test. The CAM5 simulations and the Indian Ocean pacemaker experiments display a consistent atmospheric bridge, similar to the reanalysis result.

If you added the last couple of years, does that change anything much?

Adding the last couple of years does not change the result. We use 1979–2023 ASO SST and SIC data to repeat **Supplementary Fig. 3a and 3b**, and the result is almost identical (**Fig. r7-5**).

Figure r7-5 | Regressions of SST (left; °C) and SIC (right; %) onto a normalized WTIO time series. Shown are for the ASO 1979–2023 period (without ENSO influence). Stippling indicates the 95% confidence level based on the t -test.

8. Lines 126–146: This section is easier to follow, and Fig. 2 is informative. One wonders though about the meaning or robustness of the signal, given the relatively short period of time being analysed.

Thank you for this comment. We consider your comment from the following three perspectives.

For the change of the relationship between the WTIO SST and Antarctic SIC:

(1) Observational studies on sea ice are based on the period since satellite observations in 1979. Although ERA5 data is available for the long term (i.e., from 1940 to the present), the SIC data is unreliable for the 1940–1978 periods³. Before 1979, the HadISST2 dataset was used. Specifically, prior to 1973, the SIC analysis was derived indirectly and did not include any interannual variability; for 1973–1978, the SIC was digitized by the U.S. National Climatic Data Center (NCDC) from hand-drawn U.S. National Ice Center (NIC) analyses. Thus, it gives only a general indication of sea ice

extent variations in the Southern Hemisphere on decadal timescales before the 1970s³. Fig. r8-1 shows the SIC EOF PC1 from 1940 to the present, indicating that the issues existed before 1979.

Figure r8-1 | Antarctic SIC EOF PC1 for the 1940–2020 ASO period.

Thus, the sliding correlation between the WTIO time series and SIC EOF PC1 is unachievable in the long term. But we can examine WTIO SST variability.

Antarctic SIC anomalies, acting as a “dependent variable”, are influenced by the WTIO SST variability. We use long term ERA5 SST data and add five other SST reanalysis products to examine our finding of the weakening of WTIO SST variability. These six products are:

- ERA5 (extends to long term, from 1940 to 2022) (ref⁴);
- HadISST (Hadley Centre Sea Ice and Sea Surface Temperature data set from 1940 to 2021) (ref³);
- OISST v2 (NOAA Optimum Interpolation SST version 2 from 1982 to 2021) (ref⁵);
- ORA-S5 (ECMWF Ocean Reanalysis System 5 from 1958 to 2022) (ref⁶);
- SODA3.12.2 (Simple Ocean Data Assimilation version 3.12.2 from 1980 to 2017) (ref⁷);
- GODAS (NCEP Global Ocean Data Assimilation System from 1980 to 2017) (ref⁸).

These six SST data sets show consistent WTIO 21-year sliding standard deviation results (Fig. r8-2). Focusing on the period after 1978, all of these six data sets reproduce the pronounced and coherent reduction of WTIO SST variability in recent decades, confirming the robustness of our result.

We have revised our manuscript (L133–137) and figure (Fig. 2).

Figure r8-2. A combination of six reanalysis SST products consistently shows a reduction of WTIO SST variability around 1999. The x-axis indicates the starting year in the sliding window.

(2) The multi-model mean result of CMIP6 reflects the influence of climate change (greenhouse warming).

The multi-model mean of the 21-year sliding standard deviation (i.e., models without grey shading in Fig. 5a) suggests a decrease in WTIO variability since the 1980s (Fig. r8-3; i.e., Supplementary Fig. 12). This reduction is consistent with the combination of six reanalysis products over the observation period (Fig. r8-2). The rates of decrease in WTIO SST variability in multi-model and observational data are -0.093 standard deviation decade⁻¹ and -0.208 standard deviation decade⁻¹, respectively, indicating that greenhouse warming likely contributes to the observed decrease by 45% (-0.093 divided by -0.208).

We have revised our manuscript (L266–270) and figure (Supplementary Fig. 12).

Figure r8-3 (i.e., Supplementary Fig. 12) | 21-year sliding standard deviation of WTIO time series in CMIP6. Solid blue line and shading indicate selected multi-model mean (i.e., a total of 31 out of the 45 models without gray shading in Fig. 5a) and 1.0 standard deviation of a total of 10,000 inter-realizations based on a bootstrap method for the 1979–2020 period, respectively. The x-axis indicates the starting year in the sliding window. The decline trend is -0.093 standard deviation decade⁻¹.

For the robustness of the teleconnection and SIC anomalies dominated by WTIO, we add the Indian Ocean pacemaker experiments analysis:

(3) Results from the Indian Ocean pacemaker experiments as in responses to Comment 7.

Accordingly, WTIO-induced teleconnection can be seen in a) reanalysis, b) the AGCM, and c) the Indian Ocean pacemaker experiment results. The consistency from three independent analyses confirms our finding.

We have revised our manuscript (L225–237) and figure (Fig. 4).

9. Lines 195-196: Where you say the SST anomalies in the model runs are “imposed following the WTIO SST pattern”, do you mean using the spatial patterns as shown in Figs 1 or 2, at the shown amplitude? At this point I am not clear what the WTIO SST pattern is, exactly. Is there any amplitude scaling?

Thank you for this comment.

We use the SSTA pattern in Fig. 2c (pre-1999), which represents a strong WTIO signal. And there is no amplitude scaling in forcing the AGCM experiment.

In fact, for different amplitude SST forcing, the changes in atmospheric response focused on in this paper are quasi-linear. We perform 1.66 times scaling of WTIO SSTA pattern in forcing AGCM experiment, with the equivalent amplitude to the ETIO SSTA pattern. The simulated SLP and SAT responses show similar spatial pattern to Fig. 4b, though with larger amplitudes and some regional differences (Fig. r9-1).

Figure r9-1 | Simulated SLP (Pa; upper panel), and SAT (°C; lower panel) responses to 1.66 times WTIO SST forcing (i.e., 1.66 times Fig. 2c, pre-1999). Stippling indicates the 95% confidence level based on the *t*-test.

10. Line 214: Say “...over which reliable observations are available.”

Thank you for this suggestion, and we have revised, as per your suggestion.

11. Line 271: The final statement that “melt of Antarctic sea ice could accelerate”, based on changes in the IOD teleconnection, seems unsupported to me. Yes, changes in the phenomena you identify will affect the nature of interannual variability in sea ice extent, but why would they encourage faster ice melt?

Thank you for this comment.

Over a longer period, impact of greenhouse warming is more detectable because the influence from WTIO SST variability is weaker and the climate change signal is larger^{9,10,11} relative to the “noise”, which includes interannual timescales. Thus, under greenhouse warming, melt of Antarctic sea ice could accelerate.

We have revised our manuscript (L304–306).

References:

1. Yang, D. et al. Role of tropical variability in driving decadal shifts in the Southern Hemisphere summertime eddy-driven jet. *J. Clim.* (2020).
2. Zhang, L., Han, W., Karlsruhkas, K. B., Meehl, G. A., Hu, A., Rosenbloom, N. & Shinoda T. Indian Ocean Warming Trend Reduces Pacific Warming Response to Anthropogenic Greenhouse Gases: An Interbasin Thermostat Mechanism. *Geophys. Res. Lett.* **46**, 10882-10890 (2019).
3. Rayner, N. A. et al. Global analyses of sea surface temperature, sea ice, and night marine air temperature since the late Nineteenth Century. *Journal of Geophysical Research. J. Geophys. Res.* **108**, 4407 (2003).
4. Hersbach, H. et al. The ERA5 global reanalysis. *Q. J. R. Meteorol. Soc.* **146**, 1999–2049 (2020).
5. Reynolds, R. W., Rayner, N. A., Smith, T. M., Stokes, D. C. & Wang, W. An improved in situ and satellite SST analysis for climate. *J. Clim.* **15**, 1609–1625 (2002).
6. Zuo, H., Balmaseda, M. A., Tietsche, S., Mogensen, K. & Mayer, M. The ECMWF operational ensemble reanalysis–analysis system for ocean and sea ice: a description of the system and assessment. *Ocean Sci.* **15**, 779–808 (2019).
7. Carton, J. A., Chepurin, G. A. & Chen, L. SODA3: A new ocean climate reanalysis. *J. Clim.* **31**, 6967–6983 (2018).
8. Behringer, D. & Xue, Y. Evaluation of the global ocean data assimilation system at NCEP: the Pacific Ocean. In Eighth Symposium on Integrated Observing and Assimilation Systems for Atmosphere, Oceans, and Land Surface (American Meteorological Society, 2004).
9. Kay, J. E. et al., The Community Earth System Model (CESM) large ensemble project: A community resource for studying climate change in the presence of internal climate variability. *Bull. Am. Meteorol. Soc.* **96**, 1333–1349 (2015).
10. Wallace, J. M., Deser, C., Smoliak, B. V. & Phillips, A. S. “Attribution of climate change in the presence of internal variability” in *Climate Change: Multidecadal and Beyond*, C.-P. Chang, M. Ghil, M. Latif, J. M. Wallace, Eds. (*World Scientific Series on Asia-Pacific Weather and Climate*, World Scientific, 2013), **vol. 6**, pp. 1–29.
11. Yang, K., Cai, W., Huang, G., Hu, K., Ng, B., & Wang, G. Increased variability of the western Pacific subtropical high under greenhouse warming. *Proc. Natl Acad. Sci. USA*, **119**, e2120335119 (2022).

REVIEWERS' COMMENTS

Reviewer #2 (Remarks to the Author):

I like to thank authors for considering comments and responding them in a satisfactory way. I do not have any further comments.

I may suggest following reference which could be useful.

Liu, Y., Sun, C., Li, J. et al. Decadal oscillation provides skillful multiyear predictions of Antarctic sea ice. Nat Commun 14, 8286 (2023). <https://doi.org/10.1038/s41467-023-44094-1>

Reviewer #3 (Remarks to the Author):

Thank you for the work you've done on the revised manuscript. The text reads more easily, and you have done a lot of extra work to bolster your results. In particular, the Indian Ocean pacemaker experiments are very nice and add strength to the story.

We thank the Reviewer #2 and #3 for their helpful comments.

Our responses (in blue) to reviewer's comments (in black) are as follows.

Reviewer #2 (Remarks to the Author):

I like to thank authors for considering comments and responding them in a satisfactory way. I do not have any further comments.

I may suggest following reference which could be useful.

Liu, Y., Sun, C., Li, J. et al. Decadal oscillation provides skillful multiyear predictions of Antarctic sea ice. Nat Commun 14, 8286 (2023). <https://doi.org/10.1038/s41467-023-44094-1>.

Thank you for your suggestion. We have read this reference and it is helpful to us. We have added it in the "Discussion" section of the manuscript (Line297).

Reviewer #3 (Remarks to the Author):

Thank you for the work you've done on the revised manuscript. The text reads more easily, and you have done a lot of extra work to bolster your results. In particular, the Indian Ocean pacemaker experiments are very nice and add strength to the story.

Thank you.